# Learning Likelihood-Free Reference Priors

**Nicholas Bishop** [* 1]   **Daniel Jarne Ornia** [* 1]   **Joel Dyer** [* 1]   **Anisoara Calinescu** [1]   **Michael Wooldridge** [1]

## Abstract

Simulation modeling offers a flexible approach to constructing high-fidelity synthetic representations of complex real-world systems. However, the increased complexity of such models introduces additional complications, for example when carrying out statistical inference procedures. This has motivated a large and growing literature on *likelihood-free* or *simulation-based* inference methods, which approximate (e.g., Bayesian) inference without assuming access to the simulator's intractable likelihood function. A hitherto neglected problem in the simulation-based Bayesian inference literature is the challenge of constructing minimally informative *reference priors* for complex simulation models. Such priors maximise an expected Kullback-Leibler distance from the prior to the posterior, thereby influencing posterior inferences minimally and enabling an "objective" approach to Bayesian inference that does not necessitate the incorporation of strong subjective prior beliefs. In this paper, we propose and test a selection of likelihood-free methods for learning reference priors for simulation models, using variational approximations to these priors and a variety of mutual information estimators. Our experiments demonstrate that good approximations to reference priors for simulation models are in this way attainable, providing a first step towards the development of likelihood-free objective Bayesian inference procedures.

## 1. Introduction

Simulation models have played a crucial role across diverse scientific disciplines, offering a powerful framework for understanding complex systems. From epidemiology (Kerr et al., 2021) to economics (Wiese et al., 2024; Dyer et al., 2024a) and robotics (Todorov et al., 2012), these models can be used to explore intricate dynamics, test hypotheses, and make predictions about real-world phenomena. However, these models *often lack an analytically tractable likelihood function* for the outputs they produce. This intractability prevents the direct application of classical statistical inference methods. For this reason, a set of simulation-based inference (SBI) techniques has emerged, including Approximate Bayesian Computation (Beaumont, 2019) and modern methods based on density and density ratio estimation (see, e.g., Thomas et al., 2016; Papamakarios & Murray, 2016; Hermans et al., 2020; Greenberg et al., 2019). These methods approximate common statistical inference procedures using only the ability to simulate from the simulator, rather than the ability to evaluate the model's likelihood function.

Given a simulator with implicit density $p_\theta$, a prior distribution $\pi$ over model parameters $\theta \in \Theta \subseteq \mathbb{R}^d$, and observed iid data $x_{1:n} := (x_1, \ldots, x_n)$, $x_i \in \mathcal{X}$, the goal of simulation-based Bayesian inference methods, is to approximate the posterior distribution given by Bayes's theorem:

$$\pi_{x_{1:n}}(\theta) \propto p_\theta(x_{1:n})\pi(\theta). \tag{1}$$

The prior, $\pi$, is intended to encapsulate pre-existing knowledge or beliefs about the plausible parameter values, while $\pi_{x_{1:n}}$ encapsulates the updated beliefs about plausible values for $\theta$. However, in many practical scenarios, strong prior information regarding likely values for the model parameters may be unavailable or unreliable. Even when prior beliefs exist and can be readily expressed in the form of a prior distribution, they may not be confidently held, or the modeller might fear that they are unduly influencing the results of their Bayesian analyses by incorporating strong prior beliefs into $\pi$. The modeller may therefore wish to carry out Bayesian inference in a way that minimises the influence of their own prior beliefs on the resulting inferences, and determine what posterior beliefs would arise when minimal prior information is encoded into the inference procedure.

Such considerations have motivated the development of *objective* Bayesian methods, which offer a principled approach to constructing priors that are minimally informative and, correspondingly, posteriors that are maximally data-driven. This has often been solved through reference priors (Bernardo, 1979; Berger & Yang, 1994; Bernardo, 1997;

---

[*]Equal contribution [1]University of Oxford. Correspondence to: Nicholas Bishop <nicholas.bishop@cs.ox.ac.uk>, Daniel Jarne Ornia <daniel.jarneornia@cs.ox.ac.uk>, Joel Dyer <joel.dyer@cs.ox.ac.uk>.

*Proceedings of the 42^{nd} International Conference on Machine Learning*, Vancouver, Canada. PMLR 267, 2025. Copyright 2025 by the author(s).

Berger et al., 2009), defined to be a prior distribution $\pi^*$, within some class $\Pi$ of distributions on $\Theta$, that maximizes the expected information gain from the data and minimises the influence of the prior on the posterior:

$$\pi_n^* = \arg\max_{\pi \in \Pi} \mathbb{E}_{\substack{\theta \sim \pi \\ x_{1:n} \sim p_\theta}} \left[ \log \frac{\pi_{x_{1:n}}(\theta)}{\pi(\theta)} \right]. \qquad (2)$$

The right-hand side of Equation (2) can be recognised as the mutual information $I_\pi(x_{1:n}, \theta)$ between $x_{1:n}$ and $\theta \sim \pi$. Such reference priors (RPs) have additional desirable properties, such as invariance under reparameterization, good frequentist coverage (Consonni et al., 2018), and they asymptotically achieve the minimax entropy risk when $\Pi$ is the class of continuous priors on $\Theta$ (Clarke & Barron, 1994).

Solving Equation (2) is, however, difficult except for in the simplest of cases. As described above, it is generally not the case that the model's likelihood function is tractable, let alone the mutual information (MI) between the model's parameter and output. Identifying a prior $\pi_n^*$ that solves Equation (2) for a given simulator is therefore challenging.

In this paper, we address this problem by proposing and testing a selection of likelihood-free approaches to learning RPs for implicit simulation models. A subset of these approaches allow for the simulator to be differentiable, while others address the case of non-differentiable simulators. To learn these RPs, we leverage recent advances in information theory and machine learning to estimate and maximise $I_\pi(x_{1:n}, \theta)$. Specifically, we consider both non-parametric entropy estimators, such as the Kozachenko-Leonenko estimator (Kozachenko & Leonenko, 1987; Kraskov et al., 2004), and more recent powerful neural MI estimators (Oord et al., 2018; Song & Ermon, 2020; Letizia et al., 2023). By maximizing the estimated MI, we approximate the RP without requiring explicit access to the likelihood function or the MI between $\theta$ and $x_{1:n}$. Through experiments, we demonstrate the effectiveness of our approach and compare methods across a series of benchmarks, showcasing their potential for enabling objective Bayesian inference in simulation modelling.

## 2. Background

### 2.1. Reference Priors

Reference priors (RPs) were introduced in Bernardo (1979) as an approach to designing prior distributions representing a vague initial state of knowledge. Such priors aim to be minimally informative about the parameter $\theta$, in the sense of maximising the missing information to be gained from the data. This can be seen more explicitly by rewriting Equation (2) as a difference between the prior and posterior entropies:

$$\pi_n^* = \arg\max_\pi \mathbb{H}_\pi [\theta] - \mathbb{E}_{x_{1:n} \sim m_\pi} \mathbb{H}_{\pi_{x_{1:n}}} [\theta], \qquad (3)$$

where $\mathbb{H}_\pi [\theta] = \mathbb{E}_{\theta \sim \pi} [- \log \pi(\theta)]$ is the entropy of $\theta$ when generated from $\pi$ and

$$m_\pi(x_{1:n}) = \mathbb{E}_{\theta \sim \pi} [p_\theta(x_{1:n})] \qquad (4)$$

is the marginal likelihood associated with prior $\pi$. Equation (3) makes it apparent that RPs aim to maximise the initial uncertainty in the prior while minimising, in expectation, the posterior uncertainty.

By specifying a maximally vague initial state of knowledge, RPs provide a useful tool for conducting prior sensitivity analysis: they provide the modeller with a way to assess what posterior inferences would ensue when those inferences are dominated by the observed data, rather than by the modeller's own subjective prior (Bernardo, 1997). In this way, the modeller can assess the relative influence of their own subjective prior on the final inference.

It is typical (Bernardo, 1979; Berger & Bernardo, 1992; Lafferty & Wasserman, 2001) to distinguish between the $n$-reference prior, defined by Equation (2) for finite $n$, and *the* reference prior, defined to be

$$\pi^*(\theta) = \lim_{n \to \infty} \frac{\pi_n^*(\theta)}{\pi_n^*(\theta_0)}, \qquad (5)$$

where $\theta_0 \in \Theta$ is an arbitrary reference value for $\theta$ for which $\pi_n^*(\theta_0) > 0$ for all $n$. This latter definition provides instead a measure of the *total* information that is missing about the value of $\theta$, while circumventing issues resulting from the fact that $I_\pi(x_{1:n}, \theta)$ often diverges with infinite data.

A known result in the objective[1] prior literature is that $n$-reference priors are often only supported on a finite, discrete set of atoms in $\Theta$ (Berger et al., 1988; Zhang, 1994). This restriction can be undesirable, and in conflict with the aim for a minimally informative prior. To counteract this, it is sometimes preferred to consider only *continuous, positive* priors that maximise the objective in Equation (2) (Nalisnick & Smyth, 2017); we will assume this approach when describing the methods under consideration in Section 4.

A further motivation for restricting attention to continuous priors that solve Equation (2) is provided by Bernardo (1979); Clarke & Barron (1994). Here, it is established that – asymptotically (i.e., as $n \to \infty$) and under regularity conditions (such as asymptotic Normality of the posterior distribution; see Clarke & Barron (1994) for details) – Jeffreys' prior, which has density

$$\pi_J(\theta) \propto (\det(\mathcal{F}(\theta)))^{1/2} \qquad (6)$$

---

[1]The term "objective" has been historically adopted for methods giving rise to "minimally informative" priors, according to different definitions of "minimally informative". We use this term to align with existing literature, but note that "objective" is in some senses a problematic term; see, e.g., Bernardo (1997).

is a solution, and is the unique solution among continuous positive priors. Here, $\mathcal{F}(\theta)$ is the Fisher information matrix

$$\mathcal{F}(\theta) = \mathbb{E}_{x_{1:n} \sim p_\theta} \left[ \nabla_\theta \log p_\theta(x_{1:n}) \otimes \nabla_\theta \log p_\theta(x_{1:n}) \right]$$

with determinant $\det(\mathcal{F}(\theta))$, and $\otimes$ denotes an outer product. Clarke & Barron (1994) prove that the continuous positive prior asymptotically solving Equation (2) – i.e., Jeffreys' prior, Equation (6) – induces a marginal likelihood function $m_{\pi_J}$ that asymptotically minimises the largest distance between the space $\mathcal{M}^n$ of all densities on $\mathcal{X}^n$ and members of the model family $\{p_\theta : \theta \in \Theta\}$; that is, as $n \to \infty$,

$$m_{\pi_J} = \arg \inf_{m \in \mathcal{M}^n} \left\{ \sup_{\theta \in \Theta} \mathbb{E}_{x_{1:n} \sim p_\theta} \left[ \log \frac{p_\theta(x_{1:n})}{m(x_{1:n})} \right] \right\}. \quad (7)$$

That is, asymptotically, RPs identified in the space of continuous positive distributions also satisfy the alternative notion of uninformativeness captured by (7).

### 2.2. Simulation-based Inference

Approximate, simulation-based inference (SBI) procedures for performing parameter inference for complex models have been in development for multiple decades. Beginning with approaches such as Approximate Bayesian Computation (Diggle & Gratton, 1984; Pritchard et al., 1999; Beaumont, 2019; Dyer et al., 2024b) and synthetic likelihood (Wood, 2010) methods, the past decade has seen multiple innovations in SBI methods from the machine learning community, such as through the use of neural density (e.g., Papamakarios & Murray, 2016; Greenberg et al., 2019) or density ratio (e.g., Thomas et al., 2016; Hermans et al., 2020) estimation techniques. This literature has, however, primarily focused on the problem of estimating the posterior density for a given prior and simulator, rather than on the problem of finding priors with different properties of interest. Our work addresses this gap.

## 3. Related Work

Early work on RPs have proposed procedures for pointwise estimation or generating samples via, e.g., Markov chain Monte Carlo (MCMC). An example of the former is Algorithm 1 of Berger et al. (2009), which exploits the following form of $\pi^*$ as obtained through a calculus of variations argument (Bernardo, 1979; Berger & Bernardo, 1992):

$$\pi^*(\theta) \propto \exp\left(\mathbb{E}_{x_{1:n} \sim p_\theta}\left[\log \pi_{x_{1:n}}(\theta)\right]\right). \quad (8)$$

The posterior appearing in the exponent on the right-hand side of this expression is typically taken to be the limiting posterior as $n \to \infty$, which is (under certain regularity conditions, see Van der Vaart (2000)) independent of the (continuous, positive) prior that generates it. Berger et al.

(2009) then propose to evaluate the right-hand side of Equation (8), which we denote below with $\tilde{\pi}^*$, pointwise using Mote Carlo estimates of the expectations in Equation (8) and an arbitrary positive prior $c : \Theta \to \mathbb{R}_+$:

$$\tilde{\pi}^*(\theta) \approx \exp\left(\frac{1}{R} \sum_{r=1}^{R} \log \frac{p_\theta(x_{1:n}^{(r)}) c(\theta)}{\frac{1}{L} \sum_{l=1}^{L} p_{\theta^{(l)}}(x_{1:n}^{(r)})}\right), \quad (9)$$

with $x_{1:n}^{(r)} \sim p_\theta$, $\theta^{(l)} \sim c$, and $n$ large. In contrast, Figure 1 of Lafferty & Wasserman (2001), describes an iterative MCMC algorithm inspired by the Blahut-Arimoto algorithm (Blahut, 1972; Arimoto, 1972) for sampling from a model's RP. In each of these procedures, knowledge of and the ability to easily calculate (exactly or numerically) the likelihood function corresponding to the model under consideration is assumed. More recent and more closely related work by Nalisnick & Smyth (2017) consider the problem of learning variational approximations to RPs, once again by assuming knowledge and tractability of the likelihood function associated with the model. Gao et al. (2022) consider how $n$-reference priors can be used to pre-train deep neural networks in classification settings. In contrast to these prior works, we do not assume that the modeller enjoys access to the likelihood function associated with their simulator, expanding the scope of the literature on objective priors in Bayesian inference to the case of likelihood-free simulation models that are either differentiable or non-differentiable.

## 4. Methods

In this section, we provide details on the methods we propose and test for learning ($n$-)reference priors in likelihood-free settings for simulation models. Each of the methods we consider entails optimising a (lower bound on a )n estimate $\hat{I}$ for the MI between $x_{1:n}$ and $\theta$. The main structure of the optimisation loop is shown diagrammatically in Figure 1.

Throughout, we assume the use of a parameterised variational family $\Pi := \{\pi_\phi \mid \phi \in \Phi\}$ of proper priors (see Appendix A.1) with tractable density function, such as normalising flows (Rezende & Mohamed, 2015). We will at times omit the dependence on $\phi$ for ease of notation. Such continuous approximations to ($n$-)reference priors can be advantageous, as discussed in Section 2.1. Further, by changing the number $n$ of independent replications of the simulator output, the modeller can decide whether to approximate an $n$-reference prior for smaller values of $n$, or approximate the RP (5) by taking larger $n$.

We also assume that data samples $x_{1:n}$ are generated from a simulator with implicit likelihood function $p_\theta$ that is dependent on parameters $\theta$. By sequentially sampling parameters $\theta \sim \pi$ and data $x_{1:n} \sim p_\theta$ we can construct a dataset $\mathcal{D}_\phi = (x_{1:n}^{(r)}, \theta^{(r)})_{r=1,\dots,R}$ generated from the joint distribu-

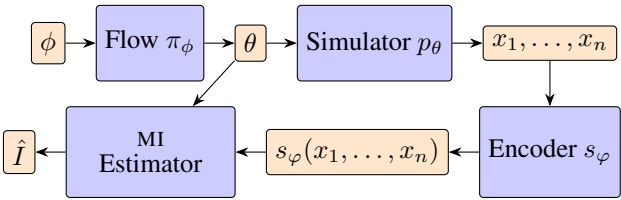

*Figure 1.* A schematic of the overall pipeline for our methods.

tion $(x_{1:n}, \theta) \sim h_\pi$, where

$$h_\pi(x_{1:n}, \theta) = p_\theta(x_{1:n})\pi(\theta) \tag{10}$$

is the joint distribution of $(x_{1:n}, \theta)$ under prior $\pi$, and discard the $\theta^{(r)}$ to produce samples from $m_\pi$.

## 4.1. Difference of Entropy Estimators

From the definition of MI between model parameters and outputs, we obtain the following equivalent expressions (see Appendix A.2):

$$I_\pi(x_{1:n}, \theta) = \mathbb{H}_\pi[\theta] - \mathbb{E}_{x_{1:n} \sim m_\pi} \mathbb{H}_{\pi_{x_{1:n}}}[\theta] \tag{11}$$

$$= \mathbb{H}_\pi[\theta] + \mathbb{H}_{m_\pi}[x_{1:n}] - \mathbb{H}_{h_\pi}[x_{1:n}, \theta]. \tag{12}$$

As one of the main approaches, we consider a set of methods that rely on estimating MI through a *difference of entropy* relation, either through estimating the entropy of the prior and the posterior (11), or through estimating the prior, marginal and joint entropies (12). The problem of estimating (and maximising) MI now becomes a problem of effectively estimating the entropy of these different distributions.

Given that we have assumed a variational prior family with tractable density $\pi(\theta)$, the first term in equations (11) and (12) can be directly estimated without bias through a Monte-Carlo entropy estimate for $B$ prior samples:

$$\hat{\mathbb{H}}_\pi[\theta] = -\frac{1}{B} \sum_{b=1}^B \log \pi(\theta^{(b)}). \tag{13}$$

Then, to estimate the terms $\mathbb{E}_{x_{1:n} \sim m_\pi} \mathbb{H}_{\pi_{x_{1:n}}}[\theta]$, $\mathbb{H}_{m_\pi}[x_{1:n}]$ and $\mathbb{H}_{h_\pi}[x_{1:n}, \theta]$ in likelihood free SBI settings there exist different *parametric* (Pichler et al., 2022) and *non-parametric* (Kraskov et al., 2004) approaches. In a non-parametric approach, we can follow Jarne Ornia et al. (2024) and exploit the differentiability of entropy estimators (e.g., the Kozachenko-Leonenko estimator, Kozachenko & Leonenko (1987)) and of differentiable simulators to propagate gradients backwards from outputs to prior; a *parametric approach* would instead construct density estimates of the corresponding distributions and compute the entropy directly. We propose the use of a parametric estimator in the interest of stability and flexibility[2].

---

[2] Non-parametric, sample based entropy estimators are usually high variance and numerically unstable (Bonachela et al., 2008).

### 4.1.1. GENERATIVE ENTROPY DIFFERENCE ESTIMATOR

Making use of (11), we devise an approach for estimating the mutual information. Recall that we can obtain estimates $\hat{\mathbb{H}}_\pi[\theta]$ directly from (13). Therefore, it only remains to find an estimate for $\mathbb{E}_{x_{1:n} \sim m_\pi} \mathbb{H}_{\pi_{x_{1:n}}}[\theta]$. Taking inspiration from Pichler et al. (2022), a practical approach is to construct parameterised estimators for the conditional distribution $\pi_{x_{1:n}}$, denoted as $\hat{\pi}_{x_{1:n}}^\psi$ (for parameters $\psi \in \Psi$), and use these to directly estimate this term using Monte Carlo sampling as in (13). Pichler et al. (2022) propose doing so by assuming the parameters $\psi$ to be a function of the conditioning variable (the observed data). This yields the estimator:

$$\hat{I}^{\psi,\phi}(\mathcal{D}_\phi) = \hat{\mathbb{H}}_\pi[\theta] + \frac{1}{B} \sum_{b=1}^B \log \hat{\pi}_{x_{1:n}^{(b)}}^\psi(\theta^{(b)}). \tag{14}$$

An important caveat is that in our case, the data marginal $m_\pi$ depends on a prior $\pi$ that changes iteratively (as the parameters $\phi$ evolve). Then, such an approach would entail repeating the following iteratively: (1) For the current prior $\pi$, sample $\mathcal{D}_\phi$ from $h_\pi$. (2) Build the estimator $\hat{\pi}_{x_{1:n}}^\psi$. (3) Compute $\hat{I}^{\psi,\phi}(\mathcal{D}_\phi)$. (4) Update $\phi$ by SGD on $\nabla_\phi \hat{I}^{\psi,\phi}(\mathcal{D}_\phi)$.

Step 2 above requires possibly learning a new estimator each time we update $\phi$, which can be sampling and computationally inefficient. Instead, we adopt a *two time-scale* approach (Borkar, 1997). We consider the prior $\pi$ and the estimator $\hat{\pi}_{x_{1:n}}$ to pertain to the same parameterised model class (particularly, a normalizing flow). We then update the conditional density estimator $\hat{\pi}_{x_{1:n}}$ with a faster learning rate[3] than the prior $\pi$.

**MI Estimator: Generative Entropy Difference (GED)** Therefore, to learn a RP using a parametric MI estimator, we propose defining $\pi$ and $\hat{\pi}_{x_{1:n}}$ to belong to some parameterised class of conditional density estimators, and use $\hat{I}^{\psi,\phi}(\mathcal{D}_\phi)$ (14) as an estimate of the MI. We then update both $\hat{\pi}_{x_{1:n}}$ at each iteration through maximum likelihood on the sampled data, and $\pi$ through SGD on the MI estimator, with a slower learning rate. For further implementation details, see Appendix B.6.

## 4.2. Variational Lower Bound Estimators

When the simulator is differentiable, we may also exploit variational lower bounds on the mutual information to learn an approximate RP. A wide range of variational lower

---

[3] The formal two time-scale argument by Borkar (1997) requires learning rates $\alpha(t)$ and $\beta(t)$ decay to zero for $t \to \infty$, to have infinite sum and finite sum of squares. Then, for the resulting stochastic approximation to guarantee convergence, $\lim_{t \to \infty} \frac{\alpha(t)}{\beta(t)} = 0$ must hold. We instead adopt an approximated argument, and choose fixed $\beta \gg \alpha$. This proved sufficient to learn the RPs correctly. See appendix for further details.

bounds have been proposed in the literature (see Poole et al. (2019) for a thorough overview). Many such bounds rely on the following variational characterization of the KL-divergence (Donsker & Varadhan, 1975) between two distributions $P$ and $Q$:

$$D_{\text{KL}}(P\|Q) = \sup_{T \in L^\infty} \mathbb{E}_P[T] - \log \mathbb{E}_Q[e^T], \quad (15)$$

where $L_\infty$ is the space of essentially bounded measurable functions. In particular, Belghazi et al. (2018) exploit Equation (15) by proposing MINE, which parameterises $T$ with a neural network (commonly referred to as a *critic*) so that a reasonably tight lower bound on the MI can be learned via stochastic gradient ascent. Due to the second term of Equation (15), which corresponds to the log-partition function of $Q$, MINE suffers from high variance, especially when the MI is large (Song & Ermon, 2020; McAllester & Stratos, 2020). To mitigate this issue, Song & Ermon (2020) propose SMILE, which clips empirical approximations of the log-parition function to lie in the range $[-\tau, \tau]$, where $\tau > 0$ is a hyperparameter controlling the bias-variance trade-off associated with truncating the log-partition function.

In the context of contrastive predictive coding, the InfoNCE objective was proposed by Oord et al. (2018) for the purpose of density ratio estimation:

$$\sup_T \mathbb{E}_{P^n(X,Y)} \left[ \frac{1}{n} \sum_{i=1}^n \log \frac{T(x_i, y_i)}{\frac{1}{n} \sum_{j \neq i} T(x_i, y_j)} \right]. \quad (16)$$

Oord et al. (2018) observes that InfoNCE implicitly maximises a variational lower bound on the MI. Song & Ermon (2020) further show that MI estimation methods derived from the Barber-Akov inequality (Barber & Agakov, 2003) may be reformulated as an optimization over density ratios. That is, from an estimate of the density ratio one may estimate MI and, by optimising a variational lower bound on the MI, one obtains an estimate of the density ratio.

Given a variational lower bound on the MI, we may jointly train a critic $T$ and a variational prior to learn a RP. The variational family is responsible for producing a prior that induces high MI between $x_{1:n}$ and $\theta$ by maximising the variational lower bound. Meanwhile, the critic is tasked with keeping the variational lower bound tight so that the variational family has a good approximation of the MI to benchmark against. Both networks may be simultaneously updated via stochastic gradient ascent. For instance, using InfoNCE, a critic $T_\mu$ with parameters $\mu$, and a flow $\pi_\phi$ we may proceed as follows:

1. Sample $\mathcal{D}_\phi \sim h_\phi$.

2. Compute the variational lower bound using critic $T_\mu$:

$$\hat{I}^{\phi,\mu}(\mathcal{D}_\phi) = \frac{1}{B} \sum_{b=1}^B \log \frac{T_\mu(x_{1:n}^{(b)}, \theta^{(b)})}{\frac{1}{B} \sum_{a \neq b} T_\mu(x_{1:n}^{(b)}, \theta^{(a)})}.$$

3. Update the parameters $\phi$ and $\mu$ via stochastic gradient ascent using $\nabla_{\phi,\mu} \hat{I}^{\phi,\mu}(\mathcal{D}_\phi)$.

The procedure outlined above relies on *differentiability* of the simulator, as gradients must backpropagate through the critic and then the simulator before reaching the variational prior. We show in experiments that this requirement, while apparently stringent, can be reliably circumvented using surrogate gradients (Blondel & Roulet, 2024).

Additionally, note that the critic and the variational prior are updated simultaneously. As a result, the critic is optimizing a non-stationary objective. If the variational prior changes too rapidly the MI estimate provided by the critic may degrade, in turn causing the performance of the variational prior to degrade. In practice, we stabilize training through a two time-scale scheme, as described in Section 4.1.

In experiments, we adopt both SMILE and InfoNCE as variational objectives, for several reasons. Many real-world simulators produce high dimensional outputs such as time series, leading to the possibility of high variance. SMILE is naturally suited to this setting due to the hyperparameter $\tau$ that enables explicit management of the bias-variance trade-off. Meanwhile, InfoNCE typically exhibits low variance, since the optimal critic does not depend on batch size (Poole et al., 2019). This property is especially important for expensive simulators, since only smaller batch sizes can be used under limited simulation budgets.

### 4.3. Performing Simulation-based Inference

It is worthwhile highlighting that, with each of the methods described in the previous subsections, the ability to perform SBI for the implicit model and learned RP $\hat{\pi}^*$ comes at no further training cost. In the case of GED, the method entails the construction of an amortised (in $x_{1:n}$) estimator $\hat{\pi}_{x_{1:n}}^\psi(\theta)$ for the posterior density $\theta$ that results from the use of the learned prior $\hat{\pi}^*$ in Bayes's theorem, which can be immediately reused to generate samples from $\hat{\pi}_{x_{1:n}}^\psi$ through, for example, Markov chain Monte Carlo (MCMC) or forward passage through the network defining the normalising flow model. Similarly, in the case of the approaches outlined in Section 4.2, which are based on optimising a variational lower bound to the mutual information between $x_{1:n}$ and $\theta$, the learned discriminators $\hat{T}$ estimate a function $w$ of the density ratio $h_{\hat{\pi}^*}(x_{1:n}, \theta)/m_{\hat{\pi}^*}(x_{1:n})\hat{\pi}^*(\theta)$. (For example, in the case of SMILE, $w$ is the identity.) Since we have assumed that the variational RP density $\hat{\pi}^*$ is tractable through the use of, for example, a normalising flow model for the variational family, estimates $\log \hat{\pi}_{x_{1:n}}^*(\theta)$ of the log-posterior density $\log \pi_{x_{1:n}}^*(\theta)$ resulting from the use of the true RP $\pi^*$ may then be obtained as

$$\log \hat{\pi}_{x_{1:n}}^*(\theta) = \log w^{-1}(\hat{T}(x_{1:n}, \theta)) + \log \hat{\pi}^*(\theta). \quad (17)$$

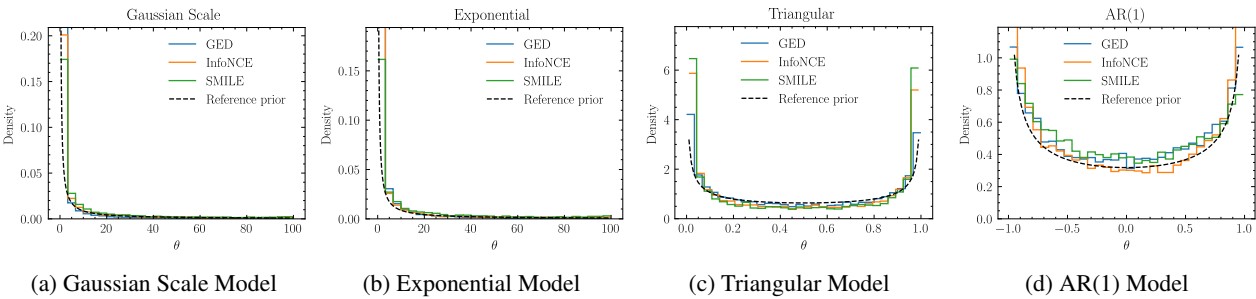

*Figure 2.* Comparison of proposed methods for learning reference priors on tractable models.

Equation (17) can then be used in, e.g., an MCMC procedure to generate samples from the posterior $\hat{\pi}^*_{x_{1:n}}$. In both cases, Bayesian SBI can immediately be performed with no further training of density ratio or posterior estimators.

## 5. Experiments

Here, we present a series of experiments[4] to assess the RP-learning methods described in Section 4. In the first instance, we consider their ability to recover the RPs for a collection of simulators with known RPs, before considering more complex simulation models whose RPs are unknown.

### 5.1. Tractable Examples

#### 5.1.1. MODEL OVERVIEWS

**Gaussian Scale Model.** The first tractable example we consider simulates Gaussian random variables. For this model, $n$ samples are generated iid from $\mathcal{N}(\mu, \sigma^2)$ as $x_t = \mu + \theta u_t$, with $u_t \sim \mathcal{N}(0, 1)$, and where $\mu \in \mathbb{R}$ is known, $\theta > 0$ is a free parameter, and $\mathcal{N}(a, b^2)$ is a Normal distribution with mean $a$ and variance $b^2$. Here, the RP (Yang & Berger, 1996) for $\theta$ is $\pi^*(\theta) \propto 1/\theta$.

**Exponential Rate Model.** We next consider an Exponential model, whose generative process is as follows: for $t = 1, \ldots, n$, we generate random variables $x_t$ from an $\text{Exp}(\theta)$ density as $x_t = -\log(1 - u_t)/\theta$, where $u_t \sim U(0, 1)$, $\theta > 0$ is a parameter, and $U(a, b)$ is a uniform distribution on $[a, b]$. As with the Gaussian scale model, the RP for $\theta$ is known (Yang & Berger, 1996) to be $\pi^*(\theta) \propto 1/\theta$.

**Triangular Model.** In this example, we generate random variables from a Triangular distribution on $[0, 1]$. Here, iid data is generated for $t = 1, \ldots, n$ as

$$
\begin{aligned}
x_t = \mathbb{I}\left[u_t \le \theta\right] \sqrt{\theta \cdot u_t} \\
+ \left(1 - \mathbb{I}\left[u_t \le \theta\right]\right)\left(1 - \sqrt{(1 - u_t)(1 - \theta)}\right),
\end{aligned} \quad (18)
$$

[4]Code available at https://github.com/joelnmdyer/lf_reference_priors.

where $\theta \in (0, 1)$ is a free parameter and $u_t \sim U(0, 1)$ is a random variable distributed uniformly on $[0, 1]$. While the RP for $\theta$ in this model is not available analytically, it is known (Berger et al., 2009) to be approximated well by a $\text{Beta}(1/2, 1/2)$ distribution. Additionally, although the derivative of $x_t$ with respect to $\theta$ for fixed $u_t$ is 0 almost everywhere when defined in the usual sense, we may nonetheless define an approximate surrogate gradient through, e.g., the straight-through gradient trick (Bengio et al., 2013) in order to backpropagate through $x_t$. In this way, we may continue to apply the methods described in Section 4.2, which require a differentiable simulator.

**Autoregressive Time-series Model.** Finally, we consider the standard autoregressive time-series model of order 1 (AR(1)). Using $u_t \sim \mathcal{N}(0, 1)$, $t = 1, \ldots, n$, this model generates a time-series $x_1, \ldots, x_n$ as

$$
x_1 = \sigma u_1, \text{ and } x_t = \theta x_{t-1} + \sigma u_t \text{ for } t = 2, \ldots, n, \quad (19)
$$

where $\sigma > 0$ is fixed and $\theta \in [-1, 1]$ is a free parameter. It can be shown (Berger & Yang, 1994) that the corresponding RP for $\theta \in [-1, 1]$ is $\pi^*(\theta) \propto \left(1 - \theta^2\right)^{-1/2}$.

#### 5.1.2. RESULTS

Figure 2 shows the obtained RPs for each method in Section 4 alongside their ground-truth counterparts. Each prior is generated from $N = 10^4$ samples and building a histogram (with fixed number of 30 bins). In general, the GED and variational lower bound (VLB) methods obtain accurate approximations of the ground truth RPs. However, bigger discrepancies can be observed especially near asymptotic parameter limits: both the Gaussian and Exponential models have asymptotic densities at $\theta = 0$, the AR(1) model has asymptotes at $\theta \in \{-1, 1\}$ and the Triangular model at $\theta \in \{0, 1\}$. These asymptotes are in general difficult to reproduce for generative models; for example, we see that InfoNCE and SMILE slightly overestimate densities close to the boundaries in Figure 2c. However, the proposed architecture of bounded generative flows are overall able to reconstruct these densities and approximate the true priors

*Table 1.* Performance metrics (mean [standard deviation] from 5 repeats) for different prior methods. **Bold** indicates best performance.

| | **Wasserstein** | | | | **C2ST** | | | |
|---|---|---|---|---|---|---|---|---|
| **Task** | Numerical | InfoNCE | SMILE | GED | Numerical | InfoNCE | SMILE | GED |
| Gauss. | 6.95 [0.64] | 7.08 [0.56] | 6.77 [0.53] | **1.66 [0.20]** | 0.90 [0.01] | 0.62 [<0.005] | 0.63 [0.01] | **0.49 [0.01]** |
| Exp. | 4.25 [0.39] | 2.43 [0.61] | 3.83 [0.75] | **2.04 [0.32]** | 0.91 [0.01] | **0.58 [0.04]** | 0.61 [0.02] | 0.59 [0.06] |
| Tri. | 0.08 [0.01] | 0.04 [<0.005] | 0.05 [0.01] | **0.02 [0.01]** | 0.64 [0.01] | 0.54 [0.02] | 0.55 [0.02] | **0.51 [0.02]** |
| AR(1) | 0.14 [0.01] | 0.06 [0.02] | **0.05 [0.03]** | 0.06 [0.02] | 0.62 [0.01] | 0.50 [0.02] | **0.50 [0.01]** | 0.51 [0.01] |

relatively accurately.

To provide a quantitative analysis, we compare each learned prior to the corresponding ground truth RP using two common metrics in the SBI literature: the Wasserstein distance and a classifier two-sample test (C2ST) (Lueckmann et al., 2021). Lower values are preferred in both cases, and indicate closer match to the ground truth RP. As a baseline, we use the numerical method described in Berger et al. (2009), which generates pointwise evaluations of the unnormalised RP using (9). Full results are provided in Table 1. From this we see that, for the Exponential, Triangular, and AR(1) models, all of our proposed methods perform consistently well. Indeed, each outperforms the numerical method from Berger et al. (2009) according to both metrics. For the Gaussian Scale model, InfoNCE marginally underperforms compared to this numerical method in terms of the Wasserstein distance to the true RP; this is likely due to the difficulties of capturing asymptotic behaviour with proper densities.

## 5.2. Complex Examples

### 5.2.1. Simple likelihood, complex posterior

We next consider the popular SBI benchmark task SLCP-D (Lueckmann et al., 2021), based on the experiment first introduced by Papamakarios et al. (2019). This simulator has 5 parameters, $\theta = (\theta_1, \ldots, \theta_5)$, taking values in $\Theta = [-3, 3]^5$ and parameterising the following data-generating process: $z_i \sim \mathcal{N}(\mu(\theta), \Sigma(\theta))$, $i = 1, \ldots, k$, where

$$\mu(\theta) = (\theta_1, \theta_2)', \quad \Sigma(\theta) = \begin{pmatrix} \theta_3^4 & \rho\,\theta_3^2\theta_4^2 \\ \rho\,\theta_3^2\theta_4^2 & \theta_4^4 \end{pmatrix} \quad (20)$$

and $\rho = \tanh\theta_5$. Distractor variables $\delta_i$ are also drawn from a mixture of Student $t$-distributions, with the final model output constructed as $x_i = (z_i, \delta_i)$. Further details are provided in Appendix C.1. The RP for $\theta$ is not analytically known. Further, this model has a higher dimensional parameter space than the experiments in Section 5.1, and is known to generate complex posteriors; this makes this experiment more challenging than those in Section 5.1.

While the RP for $\theta$ in the SLCP model is unknown, it should be expected that the RP for the transformed parameter $\vartheta = (\mu(\theta), \Sigma(\theta))$ should adhere to the shape of the RP for a multivariate Normal. In particular, we should expect the

marginal distributions for the $\mu_i(\theta)$ to be uniform in the range $[-3, 3]$, and the marginal distribution for $\det\Sigma(\theta)$, where $\det\Sigma(\theta)$ is the determinant of the model's covariance matrix, to decay rapidly in $\det\Sigma(\theta)$ (Yang & Berger, 1996).

**Results**   In Figure 3, we plot the marginal distributions for the priors learned via the InfoNCE lower bound in Section 4.2 and the GED objective in Section 4.1; we see that both methods have been able to recover this general shape, with InfoNCE doing so more successfully than GED. This is in line with our expectation that GED will underperform in this case, given that SLCP-D is known to generate complex posteriors that can be difficult to approximate accurately.

To conclude this experiment, we consider how the quality of the estimated RPs might be assessed when the form of the RP is unknown, as will be the case in practical applications. Given that the purpose of the RP is to maximise the MI between $\theta$ and model output, a simple method for assessing the relative performance of a procedure for learning RPs is to perform a classifier two-sample test. In particular, we train a binary classifier to distinguish between samples pairs $(x_{1:n}, \theta)$ drawn jointly from $h_{\pi_\phi}$ and drawn from the product of the marginal distributions, $\pi_\phi$ and $m_{\pi_\phi}$. Such metrics for assessing performance are common in SBI (see, e.g., Lueckmann et al., 2021).

In Table 2, we report classification accuracies for this classification task under four different choices for priors: a Uniform distribution on $\Theta$, and the approximate RPs estimated via InfoNCE, SMILE, and GED. From this, we see that the RPs estimated through our methods produce a higher classification accuracy than the Uniform distribution – a common default choice of prior – with the variational lower bound (VLB) methods outperforming the GED method in this instance. In summary, this practical method for assessing the quality of the learned RP corroborates our expectations that a Uniform distribution is a poor approximation to the RP for this model, while GED generates a better approximation that is inferior to those generated by the VLB methods.

### 5.2.2. An epidemic agent-based model

As a final example of a complex simulation model, we consider a susceptible-infected-recovered (SIR) agent-based model of disease spread in a population of $N$ interacting

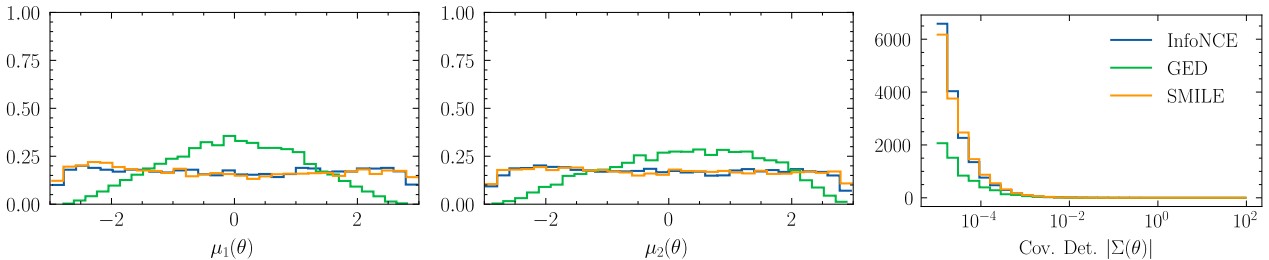

*Figure 3.* Marginal distributions of the priors learned for SLCP with distractors (SLCP-D) simulator.

*Table 2.* C2ST metrics (mean [standard deviation] from 5 repeats) for different priors in the SLCP-D experiment (Section 5.2.1). Higher values are better; **bold** denotes best performance.

| UNIFORM | INFONCE | SMILE | GED |
|---|---|---|---|
| 0.55 [0.01] | **0.99** [**<0.005**] | 0.98 [<0.005] | 0.69 [0.10] |

individuals (the "agents"). Agents in the model interact via a network, through which disease transmission occurs. Specifically, at time $t = 1, \ldots, \tau$, the state $z_{i,t}$ of agent $i$ is

$$z_{i,t} = \mathbb{I}[z_{i,t-1} = 0] \, \mathbb{I}[u_{i,t} \leq (1 - (1-\beta)^{\eta_{i,t}})]$$
$$+ \mathbb{I}[z_{i,t-1} = 1] \left(\mathbb{I}[u_{i,t} \leq \gamma] + 1\right) + 2 \, \mathbb{I}[z_{i,t-1} = 2], \quad (21)$$

where the states $z_{i,t} = 0, 1, 2$ represent, respectively, that agent $i$ is susceptible, infected, or recovered at time $t$. In the above, $u_{i,t} \sim U(0,1)$, $\eta_{i,t}$ counts the number of agent $i$'s neighbours that are infected at time $t$, and $\beta, \gamma \in (0,1)$ are parameters that determine the rate of infection and recovery. The initial states $z_{i,0} \sim \text{Bernoulli}(i_0)$, where $i_0 \in (0,1)$ determines the proportion of individuals who are infected at time 0. Further details on the model and its implementation are given in Appendix C.2. Finally, we extract the total number of infected agents at each step, such that

$$x_t = \sum_{i=1}^{N} \mathbb{I}[z_{i,t} = 1].$$

Taking $i_0 = 0.1$, $\tau = 50$, and $N = 200$, we learn an $n$-reference prior for the parameter $\theta = (\beta, \gamma) \in (0,1)^2$, where in this case $n$ is the number of iid length-$\tau$ trajectories generated at each $\theta$.

**Results** To illustrate the impact of RPs in simulation models, we plot in Figure 4 trajectories generated from the marginal likelihood functions corresponding to each of the learned priors for the SIR model. From each learned prior, we sample 10 values for $\theta$, and for each of these we generate 10 trajectories. Trajectories simulated from the same parameter value share the same colour. Visually, it is apparent that the priors that have been trained to maximise MI (via InfoNCE and GED) produce a much more diverse set of

outcomes while retaining low conditional entropy of individual parameter outcomes; in contrast, while the conditional entropies are still relatively low for the Uniform prior, the diversity in the sampled trajectories from across all parameter values is visibly lower. This can be especially useful for model designers, as with very few runs RPs produce specific (and diverse) outcomes.

Finally, we demonstrate our claim in Section 4.3 that SBI can be performed at no extra cost using the posterior and critic networks trained during GED and VLB-based methods. In particular, we compare inferences obtained by sampling directly from the posterior network trained in GED; performing MCMC using the critic network produced from the InfoNCE objective; sampling directly from a neural posterior estimator (NPE) trained as in Greenberg et al. (2019); performing MCMC using a neural density ratio estimator (NRE) trained as in Miller et al. (2022). For NPE (resp. NRE) we use the RP learned via GED (resp. InfoNCE) as the prior density, and use $10^4$ simulations. In Figure 5, we show that samples from the posterior predictive distributions for each method are comparable, demonstrating that SBI can be performed accurately at no additional training cost once RPs are learned via the GED and VLB methods we consider.

## 6. Discussion

**Main Results** In this paper, we study and test multiple approaches to learning RPs for arbitrary simulation models. Through experiments on tractable and intractable examples, we have shown that these methods can to some extent construct good RPs and attain properties of interest: they have been able to recover the overall shape of known RPs, in many cases being indistinguishable from the ground truth according to standard two-sample tests, and produce behaviour that can be seen visibly as achieving high MI between model parameters and model output in comparison to common "default" priors used in Bayesian inference (e.g., the Uniform distribution). Using these methods for constructing approximate RPs may therefore be useful in practice for complementing subjective, modeller-specified priors during

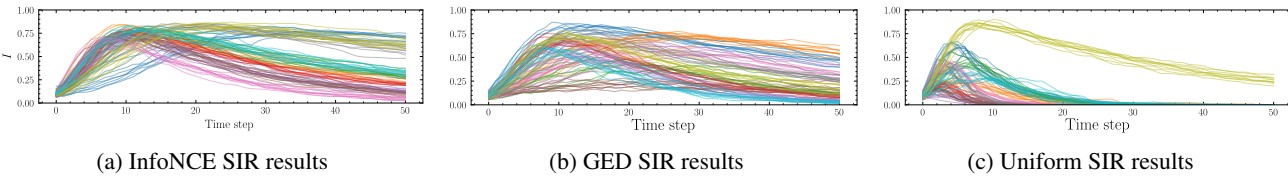

(a) InfoNCE SIR results        (b) GED SIR results        (c) Uniform SIR results

*Figure 4.* Prior predictive simulations for the SIR model, as induced by (a) the InfoNCE prior, (b) the GED prior, and (c) a Uniform prior.

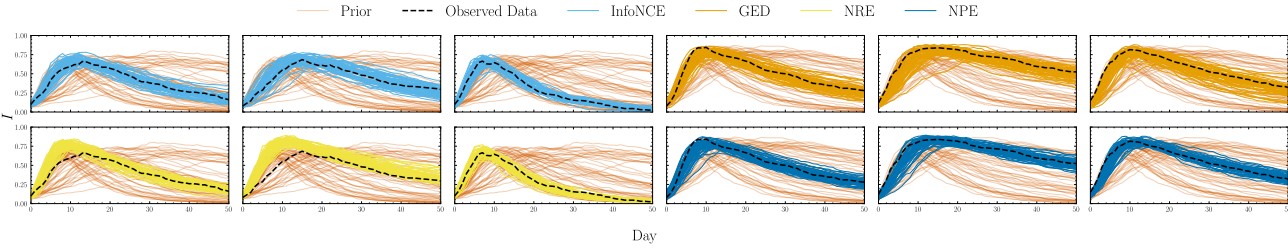

*Figure 5.* Posterior predictive samples for the SIR model conditioned on synthetic data sampled from learned RPs. The top rows displays samples generated using the density ratio/posterior estimator trained whilst learning each RP, whilst the bottom row shows samples generated using density ratio/posterior estimators seperately trained via NRE/NPE.

real-world Bayesian analyses involving simulation models: they can allow modellers to assess the degree to which their inferences differ from the case where "minimal" information – as measured by the MI between $\theta$ and data – is built in to their subjective priors. This is, we believe, the main practical benefit of our work.

**Method choice and the complexity of learning reference priors**   From a technical perspective, learning (good) RPs proved a relatively difficult task even with the chosen generative models and (deep learning) heuristics employed: RPs are not necessarily proper, which means that estimating them via generative models introduces inevitable biases and approximation errors. Further, we have on occasion seen that multiple runs of the same method can produce approximated RPs that differ substantially. This may reflect the fact that multiple (local) optima may exist; as discussed in Section 5.2.1, metrics such as classifier two-sample tests may be useful for identifying this if it occurs.

With respect to choosing between methods, a number of factors are to be considered. VLB methods seemed faster and more stable, but do not produce precise estimates of MI. Further, VLB methods involve only one density estimation task – that of estimating the prior density – while GED involves two – learning both the prior and posterior density – and for this reason VLB can be less computationally expensive and more accurate when in settings with complex posteriors. This is reflected to some degree in our experiments with the SLCP model in Section 5.2.1; however, the fact that GED has nonetheless produced a good approximate RP even in this case demonstrates that this may not be as limiting a factor. Additionally, we have seen that while GED can produce

reasonable estimates, training can also be unstable, exhibiting sensitivity to learning rates. However, GED methods can be applied to non-differentiable simulators, while currently the VLB methods we consider are only applicable to simulators for which derivatives of the model output with respect to the input parameters can be constructed through, e.g., reparameterisation tricks (see, e.g., Jang et al., 2016). Further, through GED, posterior samples can be generated rapidly via a forward pass through the posterior network, while for VLB posterior samples must be generated via MCMC. Thus, since MCMC can be more time consuming, GED methods may be preferred when fast posterior sampling is also required. In general, however, we recommend applying both methods where this can be done, since it may in general be difficult to predict beforehand which approach will perform best for any given problem.

**Limitations**   Our work naturally suffers limitations. In this paper, we focus our attention to scenarios in which all components of the parameter vector $\theta$ are of interest to the modeller, with no nuisance parameters present. Nuisance parameters introduce additional complications, and constructing RPs in their presence requires a more nuanced treatment (Bernardo, 1979; Berger & Bernardo, 1992). Finally, RPs are only one possible approach to conducting "objective" Bayesian inference, and other considerations can lead to alternative objective priors in Bayesian analysis (see, e.g., Consonni et al., 2018). The present paper is a first step towards enabling objective Bayesian inference for implicit simulation models; developing likelihood-free methods for estimating other classes of objective priors would also constitute interesting future work.

## Acknowledgements

NB, DJO, JD, AC, and MW acknowledge funding from a UKRI AI World Leading Researcher Fellowship awarded to Wooldridge (grant EP/W002949/1). MW and AC also acknowledge funding from Trustworthy AI - Integrating Learning, Optimisation and Reasoning (TAILOR), a project funded by the European Union Horizon2020 research and innovation program under Grant Agreement 952215.

## Impact Statement

This paper presents work whose goal is to advance the field of machine learning and simulation modelling. There are many potential societal consequences of our work, none which we feel must be specifically highlighted here.

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

# A. Further Background

## A.1. Proper and Improper Priors

An *improper* prior $\pi$ on space $\Theta$ is one for which the quantity

$$\int_\Theta \pi(\theta)\mathrm{d}\theta \tag{22}$$

diverges (see, e.g., Berger et al., 2009). In contrast, a prior $\pi$ is said to be *proper* if (22) is finite, such that $\pi$ corresponds to (or can be appropriately normalised to correspond to) a probability measure. While not corresponding to probability distributions in their own right, improper priors are sometimes still used in Bayesian analysis if the posterior distribution they induce via Bayes' theorem corresponds to a probability distribution; that is, if the posteriors they generate are proper.

## A.2. Relationships between Mutual Information and Entropies

We provide some basic relationships between MI and entropies for the unfamiliar reader. With MI under prior $\pi$ defined as

$$I_\pi(x_{1:n}, \theta) := \mathbb{E}_{\substack{\theta \sim \pi \\ x_{1:n} \sim p_\theta}} \left[ \log \frac{\pi_{x_{1:n}}(\theta)}{\pi(\theta)} \right], \tag{23}$$

we can separate out the logarithms to obtain

$$I_\pi(x_{1:n}, \theta) = \mathbb{E}_{\substack{\theta \sim \pi \\ x_{1:n} \sim p_\theta}} \left[ \log \pi_{x_{1:n}}(\theta) - \log \pi(\theta) \right] \tag{24}$$

$$= \mathbb{E}_{\substack{\theta \sim \pi \\ x_{1:n} \sim p_\theta}} \left[ \log \pi_{x_{1:n}}(\theta) \right] - \mathbb{E}_{\substack{\theta \sim \pi \\ x_{1:n} \sim p_\theta}} \left[ \log \pi(\theta) \right] \qquad \text{(linearity of expectation)} \tag{25}$$

$$= -\mathbb{E}_{\substack{x_{1:n} \sim m_\pi \\ \theta \sim \pi_{x_{1:n}}}} \left[ -\log \pi_{x_{1:n}}(\theta) \right] - \mathbb{E}_{\theta \sim \pi} \left[ -\log \pi(\theta) \right] \tag{26}$$

$$= -\mathbb{E}_{x_{1:n} \sim m_\pi} \left[ \mathbb{H}_{\pi_{x_{1:n}}}[\theta] \right] + \mathbb{H}_\pi[\theta], \tag{27}$$

revealing the relationship stated in (11). Finally, the relationship observed in (12) is obtained by recognising that

$$-\mathbb{E}_{x_{1:n} \sim m_\pi} \left[ \mathbb{H}_{\pi_{x_{1:n}}}[\theta] \right] = \mathbb{E}_{\substack{\theta \sim \pi \\ x_{1:n} \sim p_\theta}} \left[ \log \pi_{x_{1:n}}(\theta) \right] \tag{28}$$

$$= \mathbb{E}_{\substack{\theta \sim \pi \\ x_{1:n} \sim p_\theta}} \left[ \log \frac{h_\pi(x_{1:n}, \theta)}{m_\pi(x_{1:n})} \right] \qquad \text{(Bayes' theorem)} \tag{29}$$

$$= \mathbb{E}_{\substack{\theta \sim \pi \\ x_{1:n} \sim p_\theta}} \left[ \log h_\pi(x_{1:n}, \theta) \right] - \mathbb{E}_{\substack{\theta \sim \pi \\ x_{1:n} \sim p_\theta}} \left[ \log m_\pi(x_{1:n}) \right] \tag{30}$$

$$= -\mathbb{E}_{(x_{1:n}, \theta) \sim h_\pi} \left[ -\log h_\pi(x_{1:n}, \theta) \right] + \mathbb{E}_{x_{1:n} \sim m_\pi} \left[ -\log m_\pi(x_{1:n}) \right] \tag{31}$$

$$= -\mathbb{H}_{h_\pi}[x_{1:n}, \theta] + \mathbb{H}_{m_\pi}[x_{1:n}]. \tag{32}$$

# B. Experimental Details

In this section, we provide additional details regarding each of our experiments including the neural architectures used to implement variational families and mutual information estimators. We also provide a more in depth overview of our training procedures.

## B.1. Variational Prior Families

To describe the variational family $\Pi$ in each of our experiments we use normalizing flows. Each flow is composed of a sequence of blocks. Each block is comprised of a masked affine autoregressive layer (Papamakarios et al., 2017) and an LU permute block (Durkan et al., 2019). If required, we use a sigmoid flow layer at the end of the flow to ensure that the flow's output are bounded to lie within the range support by the simulator of study. Details about the sigmoid flow layer are presented in Section B.2.

## B.2. Sigmoid Flow Layer

The variational family $\Pi$ must be constrained to produce parameters within the range supported by the simulator. As previously mentioned, this is achieved via a sigmoid flow layer. Let $m_i$ and $M_i$ respectively define the minimum and

---

**Algorithm 1** Flow Pretraining Procedure `pretrain`

---

**Hyperparameters:** Learning rate $\beta$, Epochs $R$, Pretrain epochs $R_{\text{pre}}$
**Input:** Variational prior $\pi_\phi$
$t = 1$
**for** $t \leq R_{\text{pre}}$ **do**
    Sample $\{\theta^{(b)}\}_{b=1}^{B} \sim \pi_\phi^B$
    Compute loss $L^\phi = \frac{1}{B} \sum_{b=1}^{B} \log \pi_\phi(\theta^{(b)})$.
    Update prior parameters $\phi \leftarrow \text{ADAM}_\beta(\phi, \mu, \nabla_\phi \hat{L}^\phi)$
**end for**

---

maximum permissible values of the parameter $\theta_i$. Given an output $z \in \mathbb{R}^d$ from the previous flow layer, the sigmoid flow layer performs the following forward transformation:

$$\hat{\theta}_i = m_i + (M_i - m_i)\sigma(z_i), \quad \forall i = 1, 2, \ldots, d, \tag{33}$$

where $\sigma(z_i)$ is the logistic sigmoid function. Note that $\theta_i$ is now guaranteed to lie in the range $[m_i, M_i]$. The corresponding inverse (or backwards) transformation is given by:

$$z_i = \log(\hat{\theta}_i - m_i) - \log(M_i - \hat{\theta}_i). \tag{34}$$

Meanwhile, the log-determinants of the forward and backwards transformations can be readily computed:

$$\log \left| \det \frac{\partial \hat{\theta}}{\partial z} \right| = \sum_{i=1}^{d} \Big( \log(M_i - m_i) + \log \sigma(z_i) + \log(1 - \sigma(z_i)) \Big),$$
$$\log \left| \det \frac{\partial \hat{\theta}}{\partial z} \right| = \sum_{i=1}^{d} \Big( \log(M_i - m_i) + \log \sigma(z_i) + \log(1 - \sigma(z_i)) \Big). \tag{35}$$

### B.3. Notation for Network Architectures

Throughout the remainder of the Appendix, we will use the following notation to describe network architectures. We use $\text{MLP}(\cdot)$ to refer to fully connected multilayer perceptrons (MLPs). For example, $\text{MLP}(10, 16, 32, 64, 1)$ refers to a three layer MLP with input dimension 10, hidden layer dimensions 16, 32, and 64, and final output dimension 1. Likewise, we use $\text{LSTM}_h(\cdot)$ to refer to a stacked Long Short-Term Memory units (LSTM) with hidden state dimension $h$. For example, $\text{LSTM}_8(4, 2)$ denotes a stacked LSTM unit with input dimension 4 that is comprised of two LSTMs each with hidden dimension 8. Similarly, we use $\text{FLOW}()$ to refer to a normalizing flow. For example, $\text{FLOW}(4, 10, 16)$ refers to a normalizing flow with input dimension 4 composed of 10 blocks (as described in Section B.1) each with 16 hidden neurons. Enclosed brackets are used to denote concatenated network architectures For example $(\text{LSTM}_8(4, 1), \text{MLP}(8, 16, 1))$ denotes a network which passes a time-series of 4-dimensional inputs through an LSTM, before passing the 8-dimensional final hidden state of the LSTM to an MLP, which produces a final 1-dimensional output.

Similarly, we use $\text{SET}(\cdot, \cdot)$ to describe the set encoder architecture proposed by (Zaheer et al., 2017). More specifically, $\text{SET}(\rho, \phi)$ first applies the network $\phi$ to each individual element of a set $\{x_i\}_{i=1}^{n}$ before passing their sum to the network $\rho$, which produces the final output. As shown by Zaheer et al. (2017), such encoders are capable of representing any permutation-invariant function for sets of fixed size. As a result, set-encoders are well-suited to encoding the independently and identically distributed outputs $\{x_{1:n}^{(b)}\}$ generated by a simulator under parameter $\theta^{(b)}$. We also use $\text{TRANSFORMER}$ to refer to the transformer-based set encoder architecture proposed by (Lee et al., 2019). The transformer used is the same across all experiments and consists of two ISAB blocks and one PMA block, each with four attention heads and four inducing points.

### B.4. Pretraining Procedure for Variational Lower Bound Methods

In all of our training procedures, we first pretrain the variational prior to maximise entropy over the parameter space. Experimentally, we found that this helped prevent the reference prior from getting stuck in local optima, and reduced the

---

**Algorithm 2** Training Procedure with Variational Lower Bounds

---

**Hyperparameters:** Batch size $B$, interval $l$, Learning rates $\alpha, \beta$, epochs $R$, pretrain epochs $R_{\text{pre}}$
**Inputs:** Variational prior $\pi_\phi$, Critic $T_{\mu,\varphi}$, Simulator $p_\theta$
$\phi \leftarrow \texttt{pretrain}(\phi, \beta, R_{\text{pre}})$
$t = 1$
**for** $t \leq R$ **do**
   Sample $\{\theta^{(b)}\}_{b=1}^B \sim \pi_\phi^B$
   Simulate $x_{1:n}^{(b)} \sim p_{\theta^{(b)}}^n$ for $b = 1, \ldots, B$.
   $\mathcal{D}_\phi \leftarrow \{(\theta^{(b)}, x_{1:n}^{(b)})\}_{b=1}^B$
   Compute $\hat{I}^{\varphi,\mu}(\mathcal{D}_\phi)$ via SMILE or InfoNCE
   Update critic parameters $\mu, \varphi \leftarrow \texttt{ADAM}_\alpha((\varphi, \mu), \nabla_{\varphi,\mu}\hat{I}^{\varphi,\mu}(\mathcal{D}_\phi))$
   **if** $t \pmod l = 0$ **then**
      Update prior parameters $\phi \leftarrow \texttt{ADAM}_\beta(\phi, \nabla_\phi\hat{I}^\phi(\mathcal{D}_\phi))$
   **end if**
   $t \leftarrow t + 1$
**end for**

---

total number of epochs required for training to converge. As we use normalizing flows to implement the variational family $\Pi$ in each of our experiments, the density of the variational prior can be easily evaluated. We exploit this fact to compute a plug-in estimate of the entropy and pretrain the prior using stochastic gradient descent. This pretraining loop is described in Algorithm 1.

## B.5. Training Procedure for Variational Lower Bound Methods

Here, we outline the training procedure for learning a reference prior using a variational lower bound, such as SMILE or InfoNCE. The general procedure is outlined in Algorithm 2. On each epoch a batch of $B$ samples $\{\theta^{(b)}\}_{b=1}^B \sim \pi_\phi$ are sampled from the variational prior. Each $\theta^{(b)}$ is run on the simulator $n$ times to generate input-output pairs of the form $\mathcal{D}_\phi = \{(\theta^{(b)}, x_{1:n}^{(b)})\}_{b=1}^B$.

These pairs are used to estimate a variational lower bound on the mutual information $\hat{I}^{\varphi,\mu}(\mathcal{D}_\phi)$, which relies on a critic form $T_\varphi = f_\varphi(s_\varphi(\cdot), \cdot)$ to score input-output pairs. We refer to as $s_\varphi$ as the *encoder* as it is responsible for encoding outputs into lower-dimension representations, whilst we refer to $f_\varphi$ as the *score network* as it is responsible producing a scalar value scoring a given parameter and its corresponding encoded output. In other words, for any input-output pair $(\theta, x_{1:n})$ the score is computed as $T_{\mu,\varphi}(\theta, x_{1:n}) = f_\mu(s_\varphi(x_{1:n}), \theta)$.

As mentioned in the main body, we use two variational lower bounds in our experiments; InfoNCE and SMILE. The InfoNCE objective is given by

$$\hat{I}^{\varphi,\mu}(\mathcal{D}_\phi) = \frac{1}{B}\sum_{b=1}^B \log \frac{T_{\mu,\varphi}(x_{1:n}^{(b)}, \theta^{(b)})}{\frac{1}{B}\sum_{a \neq b} T_{\mu,\varphi}(x_{1:n}^{(b)}, \theta^{(a)})}, \tag{36}$$

while the SMILE objective is given by

$$\hat{I}^{\varphi,\mu}(\mathcal{D}_\phi) = \frac{1}{B}\sum_{b=1}^B T_{\mu,\varphi}(x_{1:n}^{(b)}, \theta^{(b)}) - \log \frac{1}{B(B-1)}\sum_{b=1}^B\sum_{a \neq b} \texttt{clip}(e^{T_{\mu,\varphi}(x_{1:n}^{(b)}, \theta^{(b)})}, e^{-\tau}, e^\tau), \tag{37}$$

where $\texttt{clip}$ denotes the clipping function $\texttt{clip}(a, b, c) = \min(\max(a, b), c)$. Given $\hat{I}^{\varphi,\mu}(\mathcal{D}_\phi)$, we then update $\phi, \varphi$ and $\mu$ using a standard gradient ascent rule such as Adam (Kingma, 2014) using $\nabla_{\phi,\varphi,\mu}\hat{I}^{\varphi,\mu}(\mathcal{D}_\phi)$. Note that the parameters $\phi$ of the variational reference prior are only updated every $l$ epochs. This provides the critic $T_{\mu,\varphi}$ time to adjust so that the variational reference prior has a good lower bound to train against. Additionally, for some complex simulators, we normalize the gradient to unit length before performing a gradient update.

The neural architectures used for each experiment are display in Table 3, whilst the training hyperparameters used are described in Table 4. The variational prior and the critic are updated with learning rates $\alpha$ and $\beta$ respectively.

*Table 3.* Neural Architectures used in each SMILE and InfoNCE experiment.

| SIMULATOR | VARIATIONAL FAMILY | ENCODER | SCORE NETWORK |
|---|---|---|---|
| GAUSSIAN SCALE | FLOW$(1,4,16)$ | TRANSFORMER | MLP$(9,64,64,64,1)$ |
| EXPONENTIAL | FLOW$(1,4,8)$ | TRANSFORMER | MLP$(2,32,1)$ |
| TRIANGULAR | FLOW$(1,5,16)$ | SET(MLP$(1,16,16,1)$, MLP$(1,8,8,1)$) | MLP$(2,16,16,1)$ |
| AR(1) | FLOW$(1,6,16)$ | (LSTM$_{16}(1,2)$, MLP$(16,1)$) | MLP$(2,8,8,1)$ |
| SLCP | FLOW$(5,6,8)$ | SET(MLP$(100,64,32)$, MLP$(32,64,32)$) | MLP$(37,64,1)$ |
| G-AND-K | FLOW$(4,8,16)$ | $\times$ | MLP$(k+1,128,1)$ |
| SIR | FLOW$(2,8,16)$ | SET((LSTM$_8(3,2)$, MLP$(8,8,16,8)$), MLP$(8,8,8,8)$) | MLP$(10,16,8,1)$ |

*Table 4.* Hyperparameter settings for InfoNCE and SMILE experiments.

| SIMULATOR | $R_{\text{PRE}}$ | EPOCHS | GRADIENT NORM | $l$ | B | $n$ | $\tau$ | $\alpha$ | $\beta$ |
|---|---|---|---|---|---|---|---|---|---|
| GAUSSIAN SCALE | 200 | 3000 | YES | 1 | 256 | 50 | 10 | $5 \times 10^{-3}$ | $5 \times 10^{-4}$ |
| EXPONENTIAL | 200 | 2000 | YES | 1 | 256 | 50 | $\infty$ | $5 \times 10^{-3}$ | $5 \times 10^{-4}$ |
| TRIANGULAR | 200 | 2000 | NO | 1 | 256 | 50 | 10 | $5 \times 10^{-3}$ | $5 \times 10^{-4}$ |
| AR(1) | 200 | 2500 | YES | 3 | 256 | 1 | 5 | $5 \times 10^{-3}$ | $5 \times 10^{-3}$ |
| SLCP | 1000 | 1000 | YES | 1 | 256 | 100 | 100 | $5 \times 10^{-3}$ | $5 \times 10^{-4}$ |
| G-AND-K | 1000 | 1000 | YES | 5 | 256 | 100 | 5 | $1 \times 10^{-3}$ | $1 \times 10^{-3}$ |
| SIR | 250 | 2000 | NO | 10 | 64 | 1 | 5 | $5 \times 10^{-3}$ | $5 \times 10^{-4}$ |

## B.6. GED

The following sections include details on our implementation of the GED method described in Section 4.1.

**Architectures** The GED method requires learning an estimator $\hat{\pi}_{x_{1:n}}^{\psi}$ for the conditional distribution $p(\theta \mid x_{1:n})$. For consistency, we use the same general architecture[5] for this estimator as for the proposal $\pi$, but we use a *conditional normal distribution* as the base distribution, and use an encoder $s_{\varphi} : X^n \to W$ that maps the sampled data $x_{1:n}$ to some latent variable $w$. As a result, the base distribution $q_0$ for the normalizing flow $\hat{\pi}_{x_{1:n}}^{\psi}$ is a conditional Gaussian denoted as $q_0(z \mid s_{\varphi}(x_{1:n}))$. The way to apply this conditional is by having $s_{\varphi}(x_{1:n}) \in \mathbb{R}^{2 \times d}$ where $d$ is the number of parameters in the model. Then,

$$q_0(z \mid s_{\varphi}(x_{1:n})) \sim \mathcal{N}(s_{\varphi}(x_{1:n})_{1:d} \mid s_{\varphi}(x_{1:n})_{d:2d}).$$

In other words, the encoder is trained to output the mean and the variance of the base flow distribution.

### B.6.1. REFERENCE PRIOR TRAINING

The main training loop follows Algorithm 4. We set $\alpha \gg \beta$ to induce a two time-scale learning dynamic. Both update steps are computed through an Adam optimizer (Kingma, 2014). As in section B.5, each parameter is simulated $n$ times to ensure that the asymptotic posterior can be well approximated. The architectures and hyperparameters used are presented in Tables 5 and 6. The same architectures were used for both the prior and the conditional density estimator per experiment.

### B.6.2. REFERENCE PRIOR TRAINING WITHOUT SIMULATOR GRADIENTS

For each experiment discussed in the main body, GED was used in conjunction with pathwise simulator gradients in order to provide a more like-for-like comparison with VLB methods. Next, we show how GED can be applied to arbitrary simulators by providing an an estimator for network gradients that does not rely on differentiability of the simulator.

To begin, we restate the objective function for GED:

$$I^{\psi,\phi}(\mathcal{D}_{\phi}) = \mathbb{E}_{\theta \sim \pi_{\phi}}[-\log \pi_{\phi}(\theta)] - \mathbb{E}_{\theta \sim \pi_{\phi}} \mathbb{E}_{x \sim p_{\theta}}[-\log \pi_{\psi}(\theta \mid x)]. \tag{38}$$

The derivative of the first term with respect to $\phi$ is easy to compute. Since the variational prior family has reparameterisable

---

[5]While we assume that the prior and posterior estimators belong to the same variational family in our experiments, in the interest of simplicity, this is not an essential aspect of this method, and in general they may belong to different variational families.

---

**Algorithm 3** Flow Pretraining Procedure `pretrain-conditional`

---

**Hyperparameters:** Learning rate $\alpha$, Pretrain epochs $R_{\text{pre}}$, Batch size $B$
**Input:** Variational prior $\pi_\phi$, Density Estimator $\hat{\pi}^\psi$
$t = 1$
**for** $t \leq R_{\text{pre}}$ **do**
    Sample $\{\theta^{(b)}, x_{1:n}^{(b)}\}_{b=1}^B \sim h_{\pi_\phi}$
    Compute loss $L^\psi = \frac{1}{B} \sum_{b=1}^B \log \hat{\pi}^\psi(\theta^{(b)} \mid s_\varphi(x_{1:n}^{(b)}))$.
    Update estimator $\psi \leftarrow \texttt{ADAM}_\beta(\psi, \nabla_\phi \hat{L}^\psi)$
**end for**

---

---

**Algorithm 4** Training for GED

---

**Hyperparameters:** Batch size $B$, Interval $l$, Learning rates $\alpha, \beta$, epochs $R$, pretrain epochs $R_{\text{pre}}$
**Input:** Variational prior $\pi_\phi$, Simulator $p_\theta$, encoder $s_\varphi$, density estimator $\hat{\pi}_{x_{1:n}}^\psi$
$\phi \leftarrow \texttt{pretrain}(\phi, \beta, R_{\text{pre}})$
$\psi \leftarrow \texttt{pretrain-conditional}(\phi, \alpha, R_{\text{pre}})$
$t = 1$
**for** $t \leq R$ **do**
    Sample $\theta^{(b)} \sim \pi$, repeat each sample $k$ times ($b = 1, \ldots, B \times k$).
    Simulate $x_{1:n}^{(b)}, \theta^{(b)} \sim h_{\pi_\phi}$, $b = 1, \ldots, B \times k$.
    **for** $1 \leq i \leq l$ **do**
        Update $\psi \leftarrow \texttt{ADAM}_\alpha \left( \psi, -\log \left( \hat{\pi}_{x_{1:n}^{(b)}}^\psi(\theta^{(b)})\} \right) \right)$
    **end for**
    Update $\phi \leftarrow \texttt{ADAM}_\beta \left( \phi, \hat{\mathbb{H}}_\pi[\theta] - \hat{\mathbb{H}}_{\pi_{x_{1:n}}}^\psi[\theta] \right)$
    $t \leftarrow t + 1$
**end for**

---

sampling paths, we have $\theta = g_\phi(z)$ where $g_\phi$ is a differentiable function (such as flow) and $z \sim t$ is a random sample from a base distribution $t$ independent of $\phi$. Using the Law of the Unconscious Statistician (Grimmett & Stirzaker, 2001), we obtain

$$\nabla_\phi \mathbb{E}_{\theta \sim \pi_\phi}[-\log \pi_\phi(\theta)] = \nabla_\phi \mathbb{E}_{z \sim t}[-\log \pi_\phi(g_\phi(z))] \tag{39}$$

$$= \mathbb{E}_{z \sim t}[-\nabla_\phi \log \pi_\phi(g_\phi(z))] \tag{40}$$

$$\approx \frac{1}{N} \sum_{n=1}^N -\nabla_\phi \log \pi_\phi(g_\phi(z^{(n)})). \tag{41}$$

Meanwhile, the derivative of the second term with respect to $\phi$ can be estimated as follows:

$$\nabla_\phi \mathbb{E}_{\theta \sim \pi_\phi} \mathbb{E}_{x \sim p_\theta} [-\log \pi_\psi(\theta \mid x)] = \mathbb{E}_{\theta \sim \pi_\phi} [\nabla_\phi \log \pi_\phi(\theta) \, \mathbb{E}_{x \sim p_\theta} [-\log \pi_\psi(\theta \mid x)]] \tag{42}$$

$$= \mathbb{E}_{\theta \sim \pi_\phi} \mathbb{E}_{x \sim p_\theta} [-\nabla_\phi \log \pi_\phi(\theta) \, \log \pi_\psi(\theta \mid x)] \tag{43}$$

$$= \frac{1}{N} \sum_{n=1}^N -\nabla_\phi \log \pi_\phi(\theta^{(n)}) \, \log \pi_\psi(\theta^{(n)} \mid x^{(n)}) \tag{44}$$

where $x^{(n)}, \theta^{(n)}$ are drawn jointly by first sampling $\theta^{(n)}$ from $\pi_\phi$, and then forward-simulating. In practice, Equation (44) can be evaluated (Paszke et al., 2019) by applying a `.stop_gradient()` method (e.g., `.detach()` in PyTorch ) to $\theta^{(n)}$ after it is sampled from the prior. Summarising, we may compute the full gradient in PyTorch as follows:

$$\frac{1}{N} \sum_{n=1}^N \left[ -\log \pi_\phi(\theta^{(n)}) + \log \pi_\phi(\theta^{(n)}.\texttt{detach()}) \cdot \log \pi_\psi(\theta^{(n)}.\texttt{detach()} \mid x^{(n)}.\texttt{detach()}) \right]. \tag{45}$$

Figure 6 shows the results of applying the above described gradient-free GED method to three of the models studied. As it can be seen quality of the learned priors differs slightly when compared to the differentiable models, but is still representative

*Table 5.* Neural Architectures Used in Each GED experiment.

| SIMULATOR | VARIATIONAL FAMILY | ENCODER |
|---|---|---|
| GAUSSIAN SCALE | $\mathrm{FLOW}(1, 8, 16)$ | $\mathrm{SET}(\mathrm{ID}, \mathrm{MLP}(64, 128, 2))$ |
| EXPONENTIAL | $\mathrm{FLOW}(1, 4, 16)$ | $\mathrm{SET}(\mathrm{ID}, \mathrm{MLP}(64, 128, 2))$ |
| TRIANGULAR | $\mathrm{FLOW}(1, 8, 16)$ | $\mathrm{SET}(\mathrm{ID}, \mathrm{MLP}(64, 128, 2))$ |
| AR(1) | $\mathrm{FLOW}(1, 8, 16)$ | $\mathrm{SET}((\mathrm{LSTM}_{32}(1, 2), \mathrm{MLP}(32, 64)), \mathrm{MLP}(64, 128, 2))$ |
| SLCP | $\mathrm{FLOW}(1, 8, 16)$ | $\mathrm{SET}(\mathrm{MLP}(100, 64, 64), \mathrm{MLP}(64, 128, 2))$ |
| G-AND-K | $\mathrm{FLOW}(4, 8, 16)$ | $\mathrm{MLP}(8, 16, 8)$ |
| SIR | $\mathrm{FLOW}(3, 8, 16)$ | $\mathrm{SET}(\mathrm{LSTM}_8(3, 2), \mathrm{MLP}(8, 16, 6))$ |

*Table 6.* Hyperparameter settings for GED.

| SIMULATOR | $R_{\mathrm{PRE}}$ | EPOCHS | GRADIENT NORM | $l$ | B | $n$ | $\alpha$ | $\beta$ |
|---|---|---|---|---|---|---|---|---|
| GAUSSIAN SCALE | 100 | 1000 | YES | 2 | 256 | 1 | $5 \times 10^{-3}$ | $1 \times 10^{-4}$ |
| EXPONENTIAL | 100 | 1000 | YES | 2 | 128 | 8 | $5 \times 10^{-3}$ | $1 \times 10^{-4}$ |
| TRIANGULAR | 100 | 1000 | YES | 2 | 256 | 8 | $5 \times 10^{-3}$ | $1 \times 10^{-4}$ |
| AR(1) | 100 | 2000 | YES | 2 | 128 | 1 | $5 \times 10^{-3}$ | $1 \times 10^{-4}$ |
| SLCP | 1000 | 3000 | YES | 4 | 256 | 100 | $5 \times 10^{-3}$ | $1 \times 10^{-4}$ |
| G-AND-K | 100 | 2000 | YES | 4 | 256 | 128 | $5 \times 10^{-3}$ | $1 \times 10^{-4}$ |
| SIR | 100 | 2000 | YES | 4 | 64 | 1 | $5 \times 10^{-3}$ | $5 \times 10^{-4}$ |

of the true underlying reference prior. As a qualitative note, training reference priors without model gradients seem to yield higher variance learning processes, which (in this case) was compensated by slightly increasing the batch sizes and learning rates.

### B.7. Entropy Estimation for Models with Tractable Likelihood functions

For simulators with tractable likelihood functions, we are able to obtain Monte Carlo estimates for the MI between $x_{1:n}$ generated from the model at $\theta$ and $\theta \sim \pi$ using the decomposition

$$I(x_{1:n}, \theta) = \mathbb{H}_{m_\pi}[x_{1:n}] - \mathbb{E}_{\theta \sim \pi} \mathbb{H}_{p_\theta}[x_{1:n}]. \tag{46}$$

Using the the Law of the Unconscious Statistician (Grimmett & Stirzaker, 2001) and the model's likelihood function, the first term in Equation (46) may be estimated as follows:

$$\mathbb{H}_{m_\pi}[x_{1:n}] = \mathbb{E}_{x_{1:n} \sim m_\pi} \left[ -\log \mathbb{E}_{\theta' \sim \pi} \left[ p_{\theta'}(x_{1:n}) \right] \right] \tag{47}$$

$$\approx -\frac{1}{B} \sum_{b=1}^{B} \log \left[ \frac{1}{R} \sum_{r=1}^{R} p_{\theta^{(r)}}(x_{1:n}^{(b)}) \right], \qquad \theta^{(r)} \sim \pi, x_{1:n}^{(b)} \sim m_\pi \tag{48}$$

This provides a (biased) estimator for $H_{m_\pi}[x_{1:n}]$. The second term can be estimated this straightforwardly as follows:

$$\mathbb{E}_{\theta \sim \pi} \mathbb{H}_{p_\theta}[x_{1:n}] = \mathbb{E}_{\theta \sim \pi} \mathbb{E}_{x_{1:n} \sim p_\theta} \left[ -\log p_\theta(x_{1:n}) \right] \tag{49}$$

$$\approx -\frac{1}{B} \sum_{b=1}^{B} \log p_{\theta^{(b)}}(x_{1:n}^{(b)}), \qquad x_{1:n}^{(b)}, \theta^{(b)} \sim h_\pi. \tag{50}$$

## C. Further Details on Complex Experiments

### C.1. Simple likelihood, complex posterior with distractors

Our implementation of SLCP with distractors follows that detailed in Appendix T.4 of (Lueckmann et al., 2021). In particular, distractors are sampled in from a mixture of $t$-distributions

$$(\delta_i)_{i=9}^{100} \sim \frac{1}{20} \sum_{j=1}^{20} t_2(\boldsymbol{\mu}^j, \boldsymbol{\Sigma}^j), \tag{51}$$

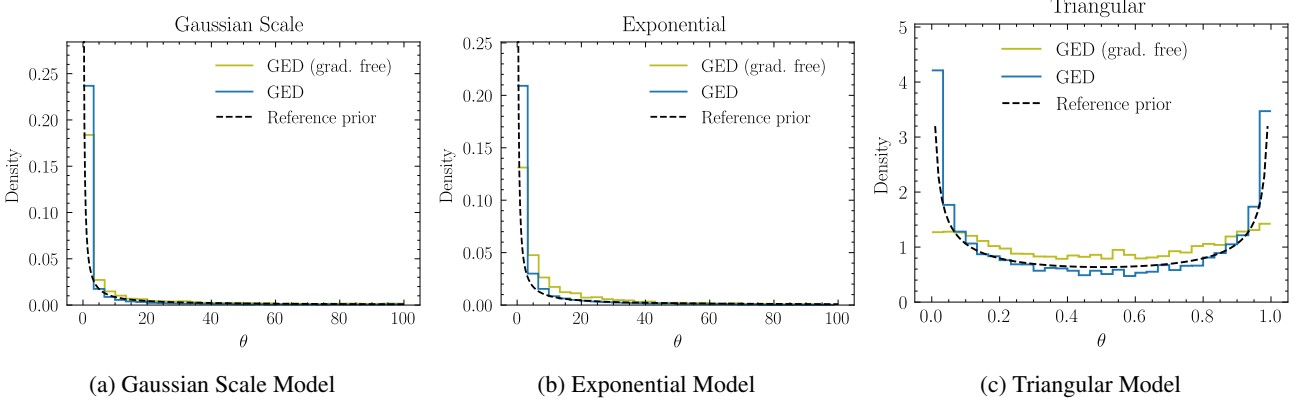

*Figure 6.* Reference prior learned for each model, with and without gradients, for the GED method.

where $\boldsymbol{\mu}^i \sim \mathcal{N}(0, 15^2\boldsymbol{I})$, $\boldsymbol{\Sigma}^i_{j,k} \sim \mathcal{N}(0, 9)$ if $j > k$, $\boldsymbol{\Sigma}_{j,j} = 3e^a$, where $a \sim \mathcal{N}(0, 1)$, and $\Sigma^i_{j,k} = 0$ otherwise.

### C.2. The SIR agent-based model

We use an SIR simulator similar in nature to the model supplied as part of the `BlackBIRDS` software package (Quera-Bofarull et al., 2023). In order to make the model differentiable, we make use of surrogate Gumbel-Softmax gradients (Jang et al., 2016) to backpropagate through Bernoulli random variables used to determine the discrete transitions between agent states. Each simulation involves simulating 200 agents for $T = 50$ time steps on a Watts-Strogatz random network (Watts & Strogatz, 1998), initialised with 10 edges per node and a rewiring probability of 0.1.

### C.3. The g-and-k model

The g-and-k model appears frequently as a benchmark case study for SBI methods (see, e.g., Fearnhead & Prangle, 2012). Inference is challenging for this model due to its ability to produce a wide range of data distributions from relatively few parameters. The generative process is as follows: for $z_t \sim \mathcal{N}(0, 1)$ and parameters $\theta = (a, b, g, k) \in \Theta = [0, 5]^4$, output $x_t \in \mathbb{R}$ is generated as

$$x_t = a + b\left(1 + c\frac{1 - \exp(-gz_t)}{1 + \exp(-gz_t)}\right)\left(1 + z_t^2\right)^k z_t\cdot, \tag{52}$$

where it is customary to take $c = 0.8$ as fixed. In this case, the true ($n$-)reference prior isn't available. However, we can expect two features to be present in the g-and-k reference prior based on the roles the parameters play in the data-generating process, (52): since $a$ plays the role of a location parameter, we can expect the reference prior to be flat in $a$ (Yang & Berger, 1996); in contrast, since $b$ is a scale parameter determining the magnitude of the contribution from the second term in (52), we might expect the reference prior to decay as approximately $1/b$ (Yang & Berger, 1996).

The RPs obtained by GED, SMILE and InfoNCE for the same random seed are plotted in Figure 7. We emphasise that each method failed to generate consistent prior estimates across random seeds, suggesting that the asymptotic RP associated with the g-and-k model is particularly difficult to estimate. We conjecture that this is due to the heavy tails of the g-and-k distribution. The development of RP estimators that perform consistently on the g-and-k model forms an interesting open challenge.

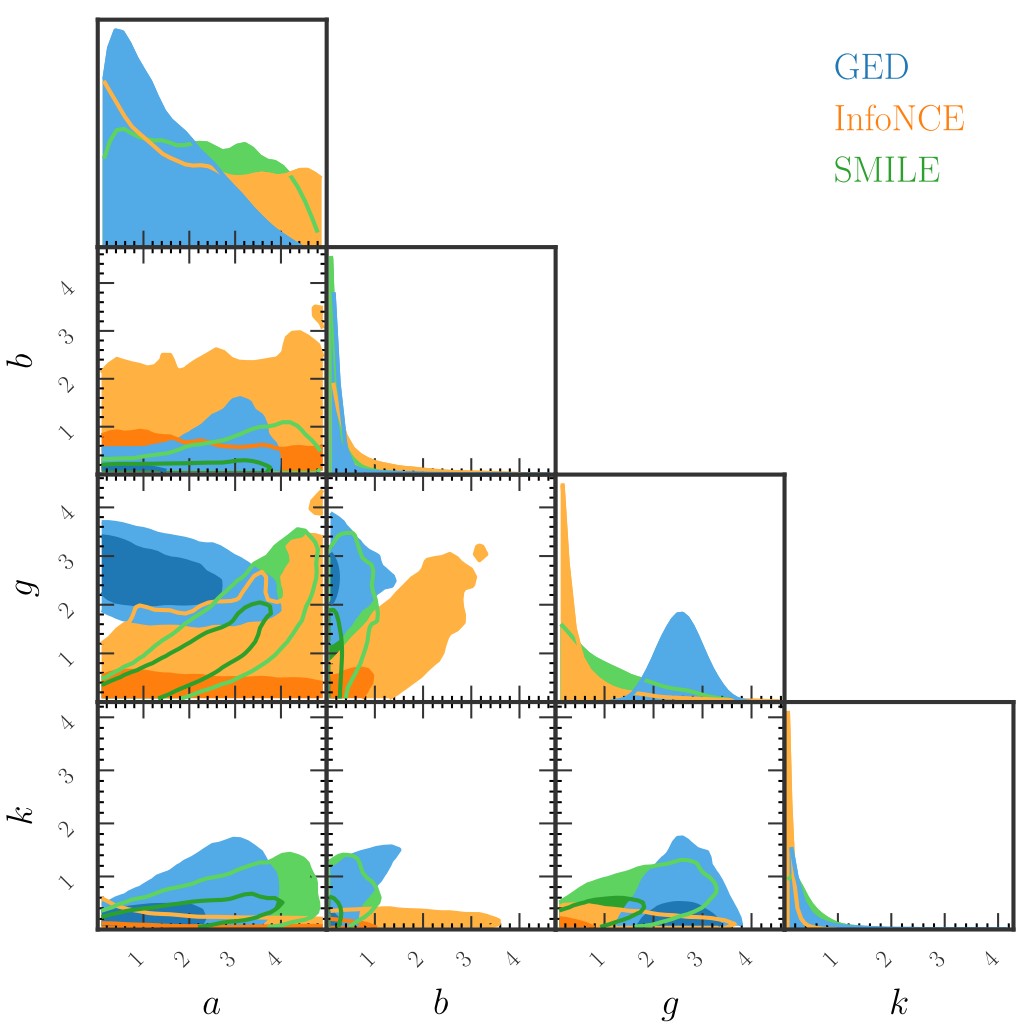

*Figure 7.* Learned RPs for the g-and-k model.

