# OpenReview forum: "Learning Likelihood-Free Reference Priors"
_ICML.cc/2025/Conference — ICML 2025 poster_

### Official Review · Reviewer_UJGD · 2025-03-04

**Overall Recommendation:** 4

**Summary:**

This paper proposes the learning of reference priors via simulation-based approaches and normalizing flows. I am very short on time for ICML reviews. Apologies for my reviews being a bit short.

**Claims And Evidence:**

They claim that they can learn reference priors via simulations. Indeed, we do see evidence that this can be done in a few simple and two more complex examples. It is a bit confusing to me that the approach is not able to learn the reference prior for two of the simple models well enough that the KS test already detects differences for small sample sizes. Did you try to learn reference priors on an unconstraint scale (e.g., log SD)?

**Essential References Not Discussed:**

Outside of reference priors, other approaches exist what use simulations to learn priors, partially even with normalizing flows. E.g., see https://arxiv.org/abs/2411.15826 , https://arxiv.org/abs/2308.11672 , https://arxiv.org/abs/2410.08710 perhaps these lines of literature should be cited too?

**Experimental Designs Or Analyses:**

Experiments are reasonable but only very low dimensional. What would happen if the normalizing flow would attempt to learn a flat reference prior over the reals, i.e. for the mean of a Gaussian model? Would that work reasonable at all?

**Methods And Evaluation Criteria:**

They seem appropriate.

**Other Comments Or Suggestions:**

nothing major

**Other Strengths And Weaknesses:**

see questions

**Questions For Authors:**

One issue I have is that, since reference priors are typically very wide, they are likely to create trouble when applied in an SBI setting to estimate the posterior. This is because, for parameters in the tails of a reference priors, simulated data is often so unreasonable that learning the posterior in the space of realistic data becomes quite hard. Can the authors comment on how applicable they see their reference priors (or reference priors more generally) in actual SBI settings beyond the few examples they provided in the paper?

**Relation To Broader Scientific Literature:**

This paper fits well into the literature on reference priors.

**Theoretical Claims:**

Only basic theory of reference priors is used which for my understanding is correctly interpreted and applied

---

> ### Author Rebuttal · Authors · 2025-04-01
>
> We thank the reviewer for their useful comments. We have included a set of new experiments, described at the end of this review.
>
> # Reviewer Comments
>
> 1. On the KS test performance: The KS tests for the Exponential and Gaussian models do reveal discrepancies. Both reference priors have asymptotes at $\theta=0$. Our failure to fully capture the shape of the reference prior results from having to constrain the output of the variational prior to a subset of the real line. Currently, the final layer of the normalising flows we use is a sigmoid layer, and it must therefore be the case that the flow's density $\to 0$ at the boundaries of the region (since it cannot assign mass to infinitely low values in logit space), meaning the current network design cannot capture an asymptote accurately. This can passing a KS test hard. Yet, aside from the asymptote, the learned and true priors are similar over the majority of the support (see updated Fig. 2 in pdf below). Note that this problem would persist even when modelling $\log(\sigma)$: either we'd leave the space unconstrained, in which case we'd fail to fully capture the uniform, improper prior on the unconstrained space, or we'd constrain to a bounded domain through a sigmoid layer again, leading to the same problem as above. We will discuss this in the revision, along with the use of alternative variational families that can better capture asymptotes.
> 2. On learning priors in unconstrained spaces and learning flat priors over the reals: Please see previous bullet point, which we hope addresses this.
> 3. On experiments being low dimensional: To demonstrate our methods' abilities to perform well for higher-dimensional parameters, we include in the pdf below (Fig. 1) new experiments on the SLCP model (description below, and further discussion in our reply to Reviewer V6ZV), which is higher-dimensional. Both methods recover expected priors; GED is less successful since it does posterior estimation, and SLCP often has complex posteriors. We'll discuss this in the revision.
> 4. On the additional references: Thank you for these suggestions. These are relevant to the problem of determining prior distributions but focus on eliciting expert priors (arguably the opposite of our problem).
> 5. On applications & practical implications of reference priors in SBI: It is possible for certain regions of parameter spaces to produce unreasonable/nonsensical outputs from simulators, and that reference priors might assign significant mass to such regions. However, the modeller can still restrict attention to regions of the parameter space $\Theta$ that do not produce absurd behaviours even when using our methods; we simply address the problem of how to learn a reference prior on a parameter space once it is given (which may be a subset of $\Theta$). We provide examples in the new experiments (Figures 2 and 3 of accompanying pdf) on doing SBI using the learned reference priors (and posteriors), and we can further point to our answer to Question 2 for reviewer RgwU, where we provide additional discussion on the use of reference priors for SBI.
>
> # New experiments
> We have included a set of new experiments here, https://github.com/ICML-7582/rebuttal_plots/blob/main/plots.pdf, including:
> - Fig. 1: Experiments on a new simulator (SLCP-D [1]) that possesses a higher dimensional parameter space. The SLCP-D model has five parameters which parameterise the mean $\mu(\theta)$ and covariance $\Sigma(\theta)$ of a 2D Gaussian. The output of SLCP-D consists of four iid samples from said Gaussian and 48 distraction vectors sampled from a mixture of Student's $t$-distributions that are independent of the parameters. SLCP-D has a relatively **s**imple **l**ikelihood that produces **c**omplex **p**osteriors. The true reference prior should be uniform in $\mu(\theta)$, and for the determinant $|\Sigma(\theta)|$ should decay rapidly in $|\Sigma(\theta)|$. VLB & GED recover this behaviour. This also highlights that our methods are approximately invariant to reparameterisation: we learn reference priors for $\theta$ but recover the correct behaviour of $\mu (\theta)$ and $|\Sigma(\theta)|$. VLB outperforms GED for SLCP-D; intuitively this is because GED approximates a (complex) posterior during training, while VLB methods do not.
> - Figs. 2 & 3: SBI experiments using reference priors from InfoNCE (a VLB method) & GED. Fig. 2: NRE posterior vs. ratio estimator trained during the VLB reference prior training; Fig. 3: NPE posterior vs. posterior density estimator trained during GED training. Inferences are almost identical, demonstrating our point about being able to perform SBI with no additional cost.
> - Fig. 4: New plots for SIR model outputs when only the infection data is used to find reference prior.
> - Fig. 5: An updated version of Figure 2 in the paper.
> - Fig. 6: An updated plot of the KS test.
>
> [1] Lueckmann et al., "Benchmarking Simulation-Based Inference", AISTATS 2021.

---

### Official Review · Reviewer_V6ZV · 2025-03-09

**Overall Recommendation:** 2

**Summary:**

This paper focuses on learning an objective prior in the context of likelihood-free inference (SBI), where the likelihood function is intractable. Within the mutual information (MI) estimation framework, the authors propose three methods: GED, InfoNCE, and SMILE. These methods are systematically compared through various toy examples in simulation studies and further applied to more complex models, including the g-and-k model and an SIR model.

**Claims And Evidence:**

This work is novel, as it may be the first attempt to obtain reference priors in the context of simulation-based inference. The paper reframes the problem of finding reference priors as a mutual information (MI) estimation task and explores solutions using various machine learning techniques. The presented simulation studies are particularly interesting, making this article an enjoyable read.

**Essential References Not Discussed:**

N/A

**Experimental Designs Or Analyses:**

1. Choice of Test Metrics:

The authors used the Kolmogorov–Smirnov two-sample test (KS test) for quantitative comparisons between the learned priors and the ground truth reference priors.

The KS test is less common and may not be ideal for assessing differences in higher-dimensional distributions or distributions with pronounced tails or boundary behaviors.

 Adopt metrics more established in SBI literature, such as classification two-sample tests (C2ST) or sliced Wasserstein distances, which can better handle high-dimensional distributions and provide clearer insights into distributional differences relevant to SBI.

2. Dimension and Complexity of Simulation Studies:

The paper primarily evaluates methods on relatively simple, low-dimensional examples. While the authors briefly consider multi-parameter models (e.g., g-and-k and SIR models), the depth of analysis for these higher-dimensional cases is somewhat limited.

Given practitioners' frequent need to handle higher-dimensional parameter spaces, it would greatly strengthen the paper to explicitly include additional high-dimensional simulation studies. Even if restricted to proper priors, this would better highlight practical applicability.

3. Stability and Consistency of Results:

There are notable inconsistencies among the GED, InfoNCE, and SMILE methods across different experiments (e.g., Section 4.2.1). While authors briefly acknowledge this, the practical implications for choosing one method over others are not adequately explored.

Provide explicit guidance or decision rules (possibly via additional experiments) on selecting among methods based on specific contexts or performance indicators.

Comparison with Standard SBI Methods:

The paper highlights that the proposed approach facilitates SBI at no extra cost but does not quantitatively compare these posterior distributions to those obtained by existing standard SBI methods.

Incorporate explicit quantitative comparisons of posterior inference accuracy or consistency with established SBI benchmarks to demonstrate added value or trade-offs.

**Methods And Evaluation Criteria:**

1. Inconsistency among the three methods

From a practical perspective, it is unclear which of the three methods should be preferred.

According to the experiments in Section 4, the GED method appears to be the most favorable. However, in Section 4.2.1, GED fails to produce a flat prior for the first parameter, $a$ whereas the other methods do. Additionally, for the parameter $g$, the support varies significantly among the methods, leading to minimal overlap in the joint distributions of GED and InfoNCE (Figure 4(d)). This discrepancy could result in entirely different Bayesian posterior inferences.

Given that the paper aims to establish an objective prior as the author illustrates in Introduction Section, these inconsistencies undermine the article’s central goal.


2. Methodological questions on GED

In Equation (12), obtaining the MI estimator requires estimating the posterior distribution \( \pi_{x_{1:n}} \), but this step is not straightforward. I believe the accuracy of posterior estimation plays a crucial role in determining the performance of the optimization in Equation (12).

Even with a parameterized prior \( \pi_\phi \), obtaining the exact posterior distribution remains challenging. Modern ML-based SBI methods \cite{papamakarios2016fast, greenberg2019automatic, lueckmann2021benchmarking} and references therein rarely achieve exact posterior estimates in many cases.

Specifically, in lines 196–198, the authors state: ``we propose defining \( \pi \) and \( \hat{\pi}_{x_{1:n}} \) to be of the same parameterized model class, and use \( \hat{I}^{\phi, \pi}(\mathcal{D}_\phi) \) (12) as an estimate of MI.” However, this explanation remains somewhat unclear. I believe this aspect should be elaborated further in the main text to clarify the underlying methodology.


3. Simulation studies

 The overall structure of the simulation studies could be improved for better clarity and organization.

 Many challenges in SBI arise from high-dimensional parameter spaces, as seen in models described in Section 4.2.1. Since multi-parameter models are a common concern among practitioners, extending the analysis beyond Section 4.1.1 to include such cases is strongly recommended, even if only proper priors are considered.

 In these cases, using classification two-sample tests (C2ST) or sliced Wasserstein distance, instead of KS statistics, would enhance consistency with the SBI literature. This would provide a more systematic comparison among the methods.

3. Minor comments and questions

 Additional derivations for clarity: The article states that Equation (2) implies (3) and that \( I_\pi(x_{1:n},\theta) \) is equivalent to Equations (9) and (10). Providing explicit derivations for these statements would improve clarity for the reader.

 Comparison with standard SBI methods (Section 3.3): This section demonstrates the ability to perform SBI at no additional cost using the learned reference prior. However, how does the posterior distribution obtained through this approach compare to those from standard SBI methods?

**Other Comments Or Suggestions:**

N/A

**Other Strengths And Weaknesses:**

N/A

**Questions For Authors:**

See above

**Relation To Broader Scientific Literature:**

This paper's key contributions lie at the intersection of simulation-based inference (SBI), objective Bayesian methods, and machine learning-based mutual information estimation.

**Theoretical Claims:**

There are not many theoretical results in this paper.

---

> ### Author Rebuttal · Authors · 2025-04-01
>
> Thank you for your helpful feedback. We have included a set of new experiments [here](https://github.com/ICML-7582/rebuttal_plots/blob/main/plots.pdf), and see reply to Reviewer UJGD for a detailed explanation.
>
> ## Methods and Evaluation Criteria
>
> - On inconsistencies between methods: It is reasonable that there may be inconsistencies between methods since there may be multiple local optima that give approximately the same mutual information between the model parameters and output but correspond to different priors. We will use some of the extra space to discuss that a sensible approach in practice may be an "ensembling" approach, namely to perform multiple runs of the prior learning procedure to account for any such variation. Such ensembling approaches are common in SBI to account for variations of this kind, see e.g. [1, 2] below.
> - On choosing between methods: Please see our discussion on this question in our response to Reviewer RgwU ('Question 3'). We will add this discussion to the final version.
> - On the performance of GED given that it learns a posterior as well as a prior: Indeed, estimating the posterior is not straightforward for some simulators. This is a possible disadvantage of GED. However, the method has produced reference priors in almost all cases matching either the ground truth or the other approaches. Other approaches might be preferable depending on problem specifications. We believe our methods (the first to address approximating reference priors in likelihood-free settings) constitute a useful baseline upon which future work can likely improve.
> - On lines 196--198: We'll use some of the extra space to give further details on the underlying methodology: we'll move the algorithm from Appendix A.6 into the main body, and specify that taking the prior and posterior estimators to be of the same model class is not necessary (the main reason for this was to reduce tuning and method complexity).
> - On higher-dimensional parameter spaces: Please see discussion on SLCP below.
> - On metrics: We have now used these metrics and present them in the tables below (format: _mean (standard dev)_ from 5 repeats). We'll use some of the extra space to discuss in the main body that we perform generally better but often similarly to/a bit worse than a likelihood-based method ("Berger") from [3].
>
> Wasserstein:
> | **Task** | **Berger** | **InfoNCE** | **SMILE** | **GED** |
> | -------- | -------- | -------- | -------- | -------- |
> | **Gaussian**     | 6.95 (0.64) | 7.08 (0.56) | 6.77 (0.53) | 1.66 (0.20) |
> | **Exponential**  | 4.25 (0.39) | 2.43 (0.61) | 3.83 (0.75) | 2.04 (0.32) |
> | **AR(1)**        | 0.14 (0.01) | 0.06 (0.02) | 0.05 (0.03) | 0.06 (0.02) |
> | **Triangular**   | 0.08 (0.01) | 0.04 (0.00) | 0.05 (0.01) | 0.02 (0.01) |
>
> C2ST:
> | **Task** | **Berger** | **InfoNCE** | **SMILE** | **GED** |
> | -------- | -------- | -------- | -------- | -------- |
> | **Gaussian**     | 0.90 (0.01) | 0.62 (0.00) | 0.63 (0.01) | 0.49 (0.01) |
> | **Exponential**  | 0.91 (0.01) | 0.58 (0.04) | 0.61 (0.02) | 0.59 (0.06) |
> | **AR(1)**        | 0.62 (0.01) | 0.50 (0.02) | 0.50 (0.01) | 0.51 (0.01) |
> | **Triangular**   | 0.64 (0.01) | 0.54 (0.02) | 0.55 (0.02) | 0.51 (0.02) |
>
> - On Equations 2, 3, 9, and 10: Thank you; we'll derive these relationships in the appendix for clarity.
> - On performing SBI: We include a demonstration of how SBI can be performed at no additional cost using our methods, and a comparison against the results of running NPE and NRE in Figs. 2 & 3 of the rebuttal pdf. We'll use the extra space to include these results in the revised paper.
>
> ## Experimental Designs or Analyses
> 1. Please see comment above on the new test metrics. For further comments and discussion on the use of the KS test, please see reply 1. to Reviewer UJGD below.
> 2. We have included new experimental results for models with a higher number of parameters (SLCP, for description see reply to UJGD). The true reference prior for the parameters should be uniform in $\mu(\theta)$ whilst the prior density for the determinant $|\Sigma(\theta)|$ should decay rapidly in $|\Sigma(\theta)|$. The priors learned by VLB and GED recover this behaviour. This highlights that our methods are approximately invariant to reparameterisation, since we learn reference priors in terms of $\theta$ but recover the correct reference prior in terms of $\mu (\theta)$ & $\Sigma(\theta)$. VLB methods outperform GED; GED approximates a (complex) posterior during training, while VLB methods don't. More generally, the tradeoffs between VLB and GED are analogous to those between NRE and NPE.
> 3. Please see our responses above on how to decide between/combine methods.
>
> [1] Cannon et al., "Investigating the impact of model misspecification in neural simulation-based inference", arXiv preprint (2022)
>
> [2] Elsemüller et al., "Sensitivity-aware amortized Bayesian inference", TMLR (2024)
>
> [3] Berger et al., "The Formal Definition of Reference Priors", The Annals of Statistics (2009)

---

> > ### Comment · Reviewer_V6ZV · 2025-04-03
> >
> > Thank you for the response. I appreciate the clarification, but I believe the concern remains only partially addressed, and in some cases, the method does not appear to perform reliably, as detailed below.
> >
> >
> >  If multiple local optima lead to priors with substantially different supports—as seen in the g-and-k example—then describing the result as “objective” becomes less convincing. As noted in the Introduction (right column, lines 39–44), the aim is to minimize the modeller’s prior influence and derive posteriors driven primarily by the likelihood. However, if the learned prior is highly sensitive to the optimization method or the method itself, this objective appears difficult to achieve in practice. While an ensemble approach may offer a practical solution, it doesn’t fully resolve the conceptual issue. A more formal justification—either theoretical or empirical—would strengthen the argument for objectivity in this framework.
> >
> > I appreciate the response regarding how to choose between methods. However, this remains unclear in practice even given the response on this concern. For instance, in the case of the g-and-k distribution, which method would be preferred? Similarly, what is the recommended approach for the SIR model? Without clear guidance or criteria, it's difficult to know how to choose between methods ahead of time.
> >
> > A key concern with InfoNCE and SMILE is their empirical performance. As shown in the simulation studies using C2ST, GED closely approximates the exact posterior, while InfoNCE and SMILE fall short—even in simple, one-dimensional settings. If these methods struggle in such basic scenarios, it raises concerns about their robustness in higher-dimensional problems. The paper would be strengthened by including C2ST results for the SLCP case (if ppssible), and by extending the evaluation to higher-dimensional examples. One straightforward approach could be to increase the dimensionality of the toy models in Section 4.1.1, where the true prior is still accessible. As another reviewer also noted, addressing performance in high-dimensional settings would significantly enhance the clarity and impact of the empirical results.
> >
> > A key concern with GED is its heavy reliance on the quality of the posterior approximation, as the authors have acknowledged. In toy examples, obtaining accurate posteriors is relatively feasible using techniques from simulation-based inference. However, in more complex settings, accurate posterior estimates are often difficult to obtain. This raises concerns not only about the objectivity of the resulting prior, but also about the method’s practical applicability—especially evident in the g-and-k distribution case. This sensitivity might also explain why InfoNCE outperforms GED on the SLCP task in the updated experiments, despite the opposite trend in the toy examples. As noted in \cite{lueckmann2021benchmarking}, SLCP tends to perform poorly when only $10^4$ simulations are used for the learning, which could be a contributing factor.
> >
> > While I recognize the potential novelty of the idea presented in the manuscript, I believe that significant revisions are necessary to bring it in line with the standards expected for publication. As such, I feel it is appropriate to maintain my original assessment score.

---

> > > ### Author Response · Authors · 2025-04-05
> > >
> > > Thank you for the detailed comment, we reply to each paragraph in your reply below.
> > >
> > > ### Para. 2
> > > The reviewer is concerned that the lack of a unique way to minimise the influence of the prior on the posterior means that approaches to doing so are not “objective”. This is a problem the reviewer has with the name “objective”, which is not our own terminology. The term “objective priors” has been historically adopted for methods giving rise to "minimally informative" priors, according to different definitions of “minimally informative”. We use this term to align with existing literature, but will discuss in our revision why “objective” is a problematic term (see, e.g., [4]).
> > >
> > > The reviewer is also concerned that, even under the formal notion of “objectivity” in Eq. 2, our approaches are not “objective” because they may result in different priors. Again, this is a problem the reviewer has with pre-existing terminology. The specific notion of “minimally informative” we consider is the mutual information (MI) between $x$ & $\theta$. This is convex in prior $\pi \in \Pi$ for fixed conditional $p_{\theta}$ (i.e., fixed simulator). Thus a unique global optimum in $\Pi$ is not guaranteed to exist and multiple ways to be “minimally informative” can exist under our operational definition of objectivity. Finding different ways to solve Eq. 2 is therefore useful for conducting a complete prior sensitivity analysis, and the fact that such solutions can substantially differ from each other isn't problematic in the way the reviewer states. As shown above and below, different metrics can be computed to check prior quality. Our revision will discuss all of this, and replace $=$ with $\in$ in Eq. 2.
> > >
> > > ### Para. 3
> > > We already give guidelines for choosing between methods (mirroring those for NPE vs NRE) in our rebuttal that we'll integrate into the revision. We'll also discuss VLB methods' stronger theoretical guarantees, which may lead practitioners to favor them over GED (e.g., InfoNCE provides a valid lower bound on MI, yielding a natural measure of "objectivity"). Even with guidelines, expecting to know ahead of time which method works best for a specific problem is unrealistic. As in any ML task, the optimal method depends on the task's details, which are often impossible to specify exactly. This is recognised in [5] (penultimate paragraph of Sec. 4), which the reviewer cites.
> > >
> > > Instead, one can test each method and compare the quality of each prior. This can be done by, e.g., using a C2ST: if a learned prior is a good reference prior (RP), it should induce high MI between $x$ & $\theta$, making it easy to classify $(x,\theta)$ pairs drawn jointly vs. pairs drawn from the product of the marginals. Higher accuracies are therefore better in this classification task. The table below gives classification accuracies for different priors for SLCP-D (_mean_ (_std_) from 5 repeats); VLB & GED methods produce RPs with high classification accuracies, and both are better RPs than a uniform prior (a typical prior for SLCP-D experiments in the SBI literature).
> > >
> > > | **Uniform** | **InfoNCE** | **SMILE** | **GED** |
> > > | -------- | -------- | -------- | -------- |
> > > |0.55 (0.01) | 0.99 (0.00) | 0.98 (0.00)  | 0.69 (0.10) |
> > >
> > > ### Para. 4
> > > Our methods -- the first likelihood-free (LF) approaches for learning RPs for arbitrary simulation models -- consistently outperform a gold standard likelihood-based approach from [3]. It's therefore difficult to see how they can be fairly described as "falling short". The order of complexity of tasks is: likelihood-based approaches to estimating RPs < LF approaches to learning RPs < LF learning of high-dimensional RPs. Our methods perform well relative to the baseline from [3] across a range of tasks in that second step, which has _no prior literature_. Thus our methods already offer a substantial improvement over the current state of the art. Further, we show above via C2STs that our learned RPs for SLCP are good, indicating high MI between $x$ and $\theta$ (the purpose of RPs) and evidencing our methods' efficacy in high dimensional settings. GED does worse comparatively, consistent with our discussion on whether to use VLB or GED for models with complex posteriors.
> > >
> > > ### Para. 5
> > > As demonstrated, GED has learned a good RP for SLCP _despite_ the known difficulties of performing posterior estimation for this model. The fact that it relies on a posterior approximation may therefore not be as limiting as it first seems. In any case, we have already discussed that VLB methods may be preferable in cases where posteriors are difficult to estimate, and we consider this insight (accompanied by the experimental results presented) a valuable contribution of our work.
> > >
> > > ### Refs
> > >
> > > [4] Irony et al., "Non-informative priors do not exist", Journal of Statistical Planning & Inference 1997
> > >
> > > [5] Lueckmann et al., "Benchmarking Simulation-Based Inference", AISTATS 2021

---

### Official Review · Reviewer_gyn2 · 2025-03-10

**Overall Recommendation:** 3

**Summary:**

The paper proposes a way to approximate a reference prior for a Bayesian analysis from a flexible family of priors in the SBI (simulation based inference) context, where the likelihood is intractable. The primary contribution here is the SBI context, in which various estimators of entropy are required to be specified. The paper uses methods adapted from those in the literature. The simulation results show that, in principle, this approach has some credibility.

**Claims And Evidence:**

Overall the principle of the method seems credible. However, there are a number of rough edges on the results, described below.

Comments on Experiments (Section 4).

It's good to see the initial focus on the known univariate reference prior examples. However the "accurate approximations of the ground truth reference priors." is slightly generous. The overall shape is there. However, in Fig 2:

* Panel (a,b) there is clearly some density estimated to exist at theta=100, which definitely should not be there. That is, there seems to be some kind of undesirable "edge effect".

* Panel (a,b) why is theta limited to 100. What happens if we consider theta larger than this? Does the method require the parameter space to be compact? How does the user know where to truncate it, if it needs to be truncated?

* Panel (c,d) there is some truncation on the y axis here. While I appreciate keeping the focus on the detail, it would be good to see just how off the approximation is on the boundaries instead of hiding it.

Fig 4:

* Panels (a-b) are claimed to have higher diversity than Panel c. Not sure about the argument being presented here. The diversity of various quantities depends on the parameterisation of the statistic being shown. Diversity in one statistic can correspond to non-diversity in a different function of the same statistic. E.g. here the top plots (statistic I) seem more diverse under the uniform prior. Though I'm sure one could compute a different statistic and have the opposite conclusion. So I'm not sure what these panels really demonstrate.

* Panel d:

- It seems quite generous to interpret the a marginal prior as uniform, under any method. Also, there seem to be finite bounds on this (unbounded) parameter. What are they?

- Its hard to judge if the marginal for b is even close to the 1/b rate as everything is so tiny. Please add the 1/b line to the plot so we can judge this more clearly. Similarly include a line on the marginals for g and k that indicate where the Gaussian case is (in which a 1/b rate might be expected). That is, the text claims that "we might expect the reference prior to decay as approximately 1/b", which may be true for a Gaussian, but is likely untrue for distributions with large skew and kurtosis.

- There are large differences among the methods. Which are we to believe? The range of differences here could be quite influential on any posterior inference.

- Similarly, what are we to make of the vastly different joint distributions? Some pairwise marginals exhibit very strong dependence, and others, very little.

Fig 2 again:

* As these models have tractable likelihoods, we can compare the outcome of the results of the current paper (the SBI focus) against other papers that have approximated reference priors with tractable likelihoods. Which of the features that are presented here are SBI-method effects, and which are not?

**Essential References Not Discussed:**

.

**Ethical Review Concerns:**

.

**Experimental Designs Or Analyses:**

See other sections.

**Methods And Evaluation Criteria:**

Because no comparisons with other (tractable likelihood) methods are performed, the authors come up with some odd ways of justifying that the results are good/credible.

* The KS tests of differences between the actual and estimated reference priors are only really a way to compare the relative performance of the different methods, in that one can pass or fail any test based on ones choice of number of samples. But then statements like (p.6. last para) "all of our proposed methods score consistently low [test statistics], avoiding the red regions where the null hypothesis is rejected." seem slightly misleading, as it seems fairly clear that if the number of samples is increased (they are mysteriously truncated at the low low value of 200 in Figure 3), then the KS statistics will fairly quickly go into the rejection region. Similarly in the discussion (p.8) the claim "in many cases [estimated priors] being indistinguishable from the ground truth according to standard two-sample tests" seems to be pushing credibility past acceptable bounds.

* As discussed in Claims and Evidence, it's not clear what point the agent-based-model example is providing. One can generate "diverse" or "non-diverse" statistics simply by choice of statistic.

**Other Comments Or Suggestions:**

Typos etc.

Abstract "uninformative reference priors". All priors are informative in some way. The authors even cite Bernardo (1997)'s paper entitled "Non-informative priors do not exist".

p.2 col 1 l.-7 "missing information to be gained" Should "missing" be "expected"? It doesn't make sense otherwise.

p.2. and elsewhere. "the Jeffreys prior" -> "Jeffreys' prior". No "the", note the punctuation. It's the prior of Mr Jeffreys: Jeffreys' prior.

p.2. Discussion of Jeffreys' prior as "a further motivation" for continuous priors. It's not clear what the narrative link to these priors is, nor why it's a further motivation.

p.3 col 1 "a (lower bound on a )n estimate". Please write better sentences that don't play such tricks.

p.6 col 2. Is "VLB" defined anywhere?

p.7 equation (18). The value 0.8 here is actually a user specified parameter (typically labelled "c"). This is how mis-understandings propagate in the literature.

p.7 last para "Early work on reference priors HAS". Also "Mote"->"Monte"

**Other Strengths And Weaknesses:**

The strengths are that the use of reference priors is something that more researchers and analysts should be using, and methods and techniques that empower them to do so, particularly in the era of SBI, are highly valuable.

The weaknesses are perhaps slightly not fully convincing simulations, and a lack of comparison to tractable-likelihood approaches to emphasis the performance of the SBI aspect more clearly.

**Questions For Authors:**

Questions are in "Claims and Evidence" and everywhere else (why are there so many sections in this review form?), and also:

* p.4, col 1, paras 2 and 3 notes that the prior and the posterior estimator are assumed to be of the same model class (so that both can be obtained by a different choice of parameters), such as a normalising flow. This is proposed for computational efficiency, but there is no discussion of the potential weaknesses or negative implications of this if, for example, the model class is unable to approximate these distributions well.

p.3 col 1. Section 3 para 2. "Continuous approximations to reference priors are advantageous for a number of reasons" we are told "as discussed in Section 2.1". Section 2.1 mentions the avoidance of a technical issue (second last paragraph), and an unclear link to Jeffrey's prior. So the merit of this claim is unclear.

**Relation To Broader Scientific Literature:**

The theoretical and methodological links to work like Nalisnick & Smith (2017) (Section 5) needed closer attention, as the contribution in this paper is purely the SBI setup.

**Theoretical Claims:**

No theoretical claims made. It's all empirical.

---

> ### Author Rebuttal · Authors · 2025-04-01
>
> Thank you for your feedback. We have included a set of new experiments [here](https://github.com/ICML-7582/rebuttal_plots/blob/main/plots.pdf), and see reply to Reviewer UJGD for a detailed explanation.
>
> ## Figure 2
> - The bumps in panels a) and b) were an artefact of the network architecture. Sigmoid layers in normalizing flows seem to induce unexpected tail behaviours in the resulting densities. Having experimented further, they have now disappeared, see Fig. 5 in the rebuttal pdf.
> - The stopping point of $\theta = 100$ was largely arbitrary, but not going higher can be justified by the fact that the prior falls off as a power law for panels a) and b).
> - Neither the reference prior theoretical foundations nor the VLB or GED methods require compact parameter sets. It happens to be that most simulators chosen require parameters to be in some compact set.
> - The question of where to truncate the prior for a potentially unbounded parameter space is important, but is a general problem when specifying priors and not specific to our methods. We instead address the problem of "given a parameter space, how could a minimally informative (proper, continuous, positive) prior be constructed?"
> - The y axes in panels c) and d) must necessarily be truncated due to the presence of asymptotes at the boundaries. We considered these plots to already adequately show that our methods have imperfections, such as InfoNCE and SMILE slightly overestimating densities close to the boundary in panel c), and GED exhibiting a slight left-skew in panel d).
> - Re. question on SBI effects: We now include a gold standard baseline for the reference priors in Fig. 2 of the accompanying pdf using the algorithm in [1] which uses knowledge of the tractable likelihood functions of these models. Plots and KS test statistics for these can be seen in Figure 6 in the rebuttal plots, and table of additional metrics requested by Reviewer V6ZV are given in our response to them below. In general we perform better, but in some cases similarly.
>
> ## Figure 4
> - We agree that diversity in one statistic $\neq$ diversity in another. We chose $x$ to be curves of the proportions of individuals who are susceptible, infected, and recovered through time. A good reference prior should -- since the MI is a difference between the entropies of the marginal likelihood $H(x)$, and the likelihood function in expectation over the prior, $H(x\mid\theta)$ -- maximise $H(x)$ (encouraging diversity _in statistic_ $x$) while minimising $H(x\mid\theta)$ (discouraging diversity _in statistic_ $x$ from individual likelihood functions, on average). Panels a) and b) in Fig. 4 aim to show this. In the accompanying rebuttal plots we include similar plots obtained by finding reference priors from GED and VLB with $x$ as the _infection curve only_. In this case, the infection curves are much more diverse.
> - Panel d): We agree that the features we expect are approximately, but not perfectly, recovered from our prior learning methods. In the revision, we will use some of the extra space to discuss reasons why we believe our methods have struggled for this g-and-k simulator. Our other experiments, in addition to our new SLCP experiments (see Fig. 1 in the rebuttal pdf), do however demonstrate that our approaches can learn useful approximations to reference priors.
>
> ## Methods and Evaluation Criteria
> Please see our discussion of our comparison to the method from [1] and about diversity above.
>
> ## Relation to Broader Literature
> We'll use some of the extra space to give more details on Nalisnick & Smith (2017) & how we differ from this prior work.
>
> ## Other Strengths and Weaknesses
> We hope our new experiments that compare the performance of our methods to the gold standard tractable-likelihood approach from [1] (see above & response to Reviewer V6ZV) and demonstrate the ability of VLB & GED to perform SBI accurately at no extra cost (see Fig. 2 & 3 in accompanying pdf) address your concerns.
>
> ## Other Comments
> We'll fix the typos. We'll also: say "minimally informative", not "uninformative", in abstract; use "Jeffreys' prior"; write "optimise an estimate of, or a lower bound on, the MI..."; ensure "VLB" is defined on first use; and write $c$ in Eq. 18 and state that $c=0.8$ is a common choice that we also use. We'll explain the (missing) narrative link to Jeffreys' prior; the point of this, and Eq. 7 in particular, was to highlight that continuous, positive distributions satisfying Eq. 2 also (asymptotically & under regularity conditions) satisfy an alternative notion of "minimally informative" given in Eq. 7. Finally, we'll state these regularity conditions.
>
> ## Questions
> Our use of a prior and posterior estimator that are of the same model class is just for ease of hyperparameter tuning, and not a requirement of GED. On the comment re. continuous approximations: Please see "Other comments" above.
>
> [1] Berger et al., "The Formal Definition of Reference Priors", The Annals of Statistics (2009)

---

### Official Review · Reviewer_RgwU · 2025-03-12

**Overall Recommendation:** 4

**Summary:**

This paper addresses the problem of constructing reference priors for simulation-based
inference (SBI). Unlike most SBI research, which focuses on posterior or likelihood
estimation given a user-defined prior, this work tackles the challenge of developing
"uninformative" or "reference" priors in a principled way when strong prior knowledge is
unavailable or undesirable.  The authors formalize the problem in the context of SBI,
where the likelihood is intractable, and propose several approaches for learning
reference priors using normalizing flows. These methods, adapted from the existing
reference prior literature, are based on variational approximations and mutual information
estimators. The paper demonstrates and compares these methods on several benchmark
tasks, including toy examples with analytically tractable reference priors and more
complex, intractable examples. The goal is to enable "objective" Bayesian inference in
SBI, minimizing the influence of subjective prior beliefs on the posterior.

## Update After Rebuttal
I thank the authors for their detailed rebuttal and the inclusion of additional experiments. This effectively addressed all my questions and concerns. With the promised changes incorporated, I believe the paper represents a valuable contribution to the field, consistent with my initial positive evaluation.

**Claims And Evidence:**

The primary claim – that reference priors can be learned in the SBI context using the
proposed methods – is well-supported. The paper provides clear derivations of the
relevant algorithms, adapting them from established reference prior literature.  The
experiments on various benchmark tasks demonstrate the feasibility of learning
approximations to reference priors.  The theoretical grounding is solid.

A secondary, implicit claim is that learning reference priors with these methods enables
standard SBI via subsequent posterior estimation or density-ratio estimation, followed
by MCMC sampling. While this is logically sound (given the learned prior, which provides
either a likelihood estimate or a posterior/ratio estimate), the paper lacks a
comprehensive empirical demonstration of this end-to-end practical application. While
the authors briefly mention a comparison to the uniform prior predictive in the SIR
example, this is insufficient to fully showcase the utility.  What's missing are
experiments that systematically compare posterior inference (including both posterior
distributions and predictive performance) using a learned reference prior versus a
standard, hand-crafted prior (e.g., uniform or a domain-informed prior). These
comparisons should highlight the practical advantages, if any, of using the learned
reference prior in subsequent SBI tasks. Crucially, this demonstration should emphasize
that once the reference prior is learned, it can be readily reused for standard SBI
without requiring further training, which is a key potential benefit of the proposed
approach. This expanded demonstration would significantly strengthen the practical
justification for the methods. This is not a major flaw, but a missed opportunity.

**Essential References Not Discussed:**

The paper provides a good discussion of prior work on learning reference priors. A
connection to generalized Bayesian inference (or "post-Bayesian" inference) could
strengthen the contextualization.  This related field, with applications in SBI (e.g.,
Matsubara et al., 2021; Gao et al., 2023; Jävernpää et al., 2025), offers alternative
approaches to handling uncertainty and model misspecification. While this paper focuses
on a different approach (the prior), discussing the relationship and contrasting it with
generalized Bayesian inference would provide a more complete view of the research
landscape.

**Experimental Designs Or Analyses:**

Yes, I reviewed the experimental design and analyses, and they appear sound in their current form (apart from the additional experiments and demonstrations mentioned below).

**Methods And Evaluation Criteria:**

Yes, the experimental design is generally appropriate. Using toy examples with
analytically tractable reference priors provides a crucial validation of the learned
priors.  The more complex examples add further evidence, although a more in-depth
discussion of the choices made, or the use of a well-established, real-world SBI
benchmark (e.g., from a field like neuroscience or astrophysics, where SBI is commonly
applied), would increase the practical relevance and impact.

**Other Comments Or Suggestions:**

1) line 198: "update update"

**Other Strengths And Weaknesses:**

### Strengths

- The paper is generally very well-written and clearly structured.
- The problem and motivation are clearly articulated, and the necessary background is
  introduced thoroughly yet concisely.
- The contributions are clearly stated.

### Weaknesses/Suggestions

- Concept Figure: Figure 1 is quite minimal and sketch-like. A more prominent and
  informative figure, appearing earlier in the paper, would help convey the overall
  approach more effectively.
- Practical Context for SBI: While the paper argues for the use of reference priors, it
  would benefit from a more thorough discussion of the practical context within SBI. The
  introduction mentions that the modeler might want to minimize the influence of their
  prior beliefs. However, in many practical SBI applications, practitioners do have
  prior constraints (e.g., bounds on interpretable simulator parameters). This
  difference between the "fully objective Bayesian" ideal and the typical SBI use-case
  should be explicitly addressed.

**Questions For Authors:**

1) Proper vs. Improper Priors: On page 6, the paper mentions that the Exponential and
   Scale Gaussian models have improper reference priors. However, this distinction
   between proper and improper priors wasn't introduced earlier in the paper. Could you
   clarify this point and perhaps provide a brief discussion of the implications of
   using improper priors in this context?
2) Practical Application: Could you elaborate on how the proposed methods would be
   applied in a practical SBI scenario? For instance, consider a well-known SBI
   benchmark like the Hodgkin-Huxley model, which is often used with uniform priors.
   - How would one design and learn a reference prior in this case?
   - What are the expected computational costs compared to using a uniform prior?
   - How would a practitioner evaluate whether the learned reference prior is "good" or "sensible"?
   - What would be the conceptual and practical benefits for the practitioner?
3) Method Selection: The paper proposes several methods for learning the reference prior
   (e.g., VBEM, direct optimization, sequential learning). How should a practitioner
   choose among these options in a given application? Are there any heuristics or
   guidelines that can inform this choice, based on factors like the complexity of the
   simulator, the dimensionality of the parameter space, or the computational resources
   available? If no such heuristics currently exist, could you outline a potential
   research direction for developing them?

**Relation To Broader Scientific Literature:**

The paper makes a valuable contribution by addressing a largely unexplored area within
SBI: the principled construction of reference priors. While reference priors are
well-established in Bayesian inference, their application to SBI, with its intractable
likelihoods, is novel.  This work connects to the ongoing discussion about the
faithfulness of SBI methods (e.g., Hermans et al., 2023, concerning posterior
calibration and overconfidence) by highlighting the often-overlooked role of the prior
in the overall inference pipeline. It addresses the need for "objective" Bayesian
inference in SBI, where minimal prior information is incorporated.

**Theoretical Claims:**

I did not check the proofs in detail, focusing on the conceptual soundness and experimental validation.

---

> ### Author Rebuttal · Authors · 2025-04-01
>
> Thank you for the valuable feedback. We have included a set of new experiments, see the reply to Reviewer UJGD for a detailed explanation.
>
> ## Questions
>
> 1. We will include explicit technical definitions for, and a discussion of, proper and improper priors. Briefly: an improper prior is an "unnormalisable" prior, i.e. one with infinite integral over the support, while a proper prior can be normalised to have integral 1; improper priors do not strictly correspond to probability measures for this reason, but can sometimes still produce proper Bayesian posterior distributions.
> 2. We will use some of the additional space to include a discussion on these 4 points at the end of our revised paper. Briefly:
>     - The first of these points we discuss in bullet point 3. directly below this one.
>     - Using a user-specified (e.g., Uniform) prior may be cheaper since it involves no learning of a prior; however, we are not recommending that our learned reference priors replace any subjective priors the modeller believes in, but rather that they complement such subjective priors by enabling a prior sensitivity analyses. Our methods allow modellers to demonstrate the degree to which their inferences differ from the case where "minimal" information (as measured by the MI between $\theta$ and $x$) is built in to the prior, and so we envision that our methods generate priors that are used _alongside_ the modeller's subjective prior.
>     - To assess whether a reference prior is sensible (whether it serves the purpose it intends to serve) the modeller can estimate the MI between $\theta$ and $x$ using the different networks trained in the VLB and GED methods, or e.g. via sample-based entropy estimators [1]. However, note that estimating MI from finite data is a challenging problem (see [2], [3]). Alternatively, one can measure divergences between prior and posterior distributions as an estimate of the MI to assess these.
>     - We outline what we believe is the main practical and conceptual benefit in the second bullet point above, and will further emphasise this in the revision.
> ### Question 3: Differences between VLB and GED
> We will include detailed heuristics in the paper. VLB methods maximise a variational lower bound for the MI which relies on learning critic networks to (essentially) construct density ratio estimates; GED methods construct density estimates from samples of the marginal and conditional distributions of $(\theta, x)$ and maximise directly the estimated MI.
> - GED methods do not require differentiability so they are preferred for problems with non-differentiable data collection, but (for equally computationally complex simulators) VLB methods are less computationally expensive than GED methods (in the sense that they only need to learn one density estimator instead of two).
> - GED methods rely on constructing an estimate for $\pi(\theta \mid x)$. In problems prone to yield very complex posterior distributions, this may hinder the efficacy of the method. In these, VLB methods might be preferred (when applicable), since this entails only one density estimation task on $\Theta$ (i.e., learning the prior), while GED involves two (i.e., both the prior and posterior).
> - When successfully learning a reference prior $\pi$ via GED, we also get (at no extra cost) an estimator for the posterior $\pi(\theta \mid x)$ under $\pi$, which we can directly use for e.g. SBI. We did this in the new rebuttal experiments (see Figures 2 and 3).
>
> ## Weaknesses
> 1. We will add detail to Figure 1 in the revision for clarity and place it earlier.
> 2. We agree that practitioners will often have prior constraints, and we support the incorporation of such beliefs into their Bayesian analysis. As we discuss above, we do not argue against the use of such subjective priors, but instead intend to provide practitioners with tools to assess how much information they have built into their prior, by equipping them with methods that let them find "minimally informative" priors. Our methods can of course also be used when no strong subjective prior beliefs exist, or in situations in which the practitioner prefers to minimise their own influence on the Bayesian analysis for whatever reason.
>
> ## General comments
>
> - **On the connection to GBI**: We’ll use some of the extra space to briefly discuss connections to GBI.
> - **Line 198**: Thank you, we'll correct this typo.
>
> ## References
> [1] Kraskov, Alexander, Harald Stögbauer, and Peter Grassberger. "Estimating mutual information." Physical Review E—Statistical, Nonlinear, and Soft Matter Physics 69.6 (2004): 066138.
>
> [2] Song, Jiaming, and Stefano Ermon. "Understanding the limitations of variational mutual information estimators." arXiv preprint arXiv:1910.06222 (2019).
>
> [3] McAllester, David, and Karl Stratos. "Formal limitations on the measurement of mutual information." International Conference on Artificial Intelligence and Statistics. PMLR, 2020.

---

> > ### Comment · Reviewer_RgwU · 2025-04-08
> >
> > I thank the authors for their detailed rebuttal and additional experiments. All my concerns and questions have been addressed. With the inclusion of the additional results and promised changes, this paper will make a valuable contribution to the SBI literature, which is consistent with my initial positive evaluation. I look forward to seeing the final version.

---

> > > ### Author Response · Authors · 2025-04-09
> > >
> > > It's great to hear that the reviewer's concerns have been addressed and we thank them again for their helpful comments.

---

### Decision · Program_Chairs · 2025-05-01

**Decision:**

Accept (poster)

**Comment:**

This paper studies the topic of simulation-based inference, which has the challenge of specifying a prior. This paper considers the problem of specifying an uninformative reference prior, and proposes and tests likelihood-free methods for learning reference priors by maximizing mutual information between parameters and simulated data, using variational approximations and both classical and neural mutual information estimators.

Reviewers were overall positive about this paper. There has not been much work on constructing reference priors in the SBI literature, and so reviewers thought this was a valuable contribution to the literature that lays the groundwork for objective Bayes in SBI. There were some initial concerns about the experimental results, including baselines. The authors provided a detailed rebuttal that included additional experiments. Several reviewers said that the promised changes in the rebuttal addressed their primary concerns.